



# A climatology of atmospheric rivers over Scandinavia and related precipitation

Erik Holmgren[1] and Hans W. Chen[1]

[1]Department of Space, Earth and Environment, Chalmers University of Technology, Gothenburg, Sweden

**Correspondence:** Hans W. Chen (hans.chen@chalmers.se)

**Abstract.** Atmospheric rivers (ARs) play an important role in the global climate system, facilitating both meridional moisture transport and regional weather patterns that are important for the local water supply. While previous research has mainly focused on the relationship between ARs and precipitation in North America and East Asia, the role of ARs in the regional climate of Scandinavia remains understudied. In this study, we used data from the Atmospheric River Tracking Method Inter-comparison Project to characterize ARs making landfall in Scandinavia during 1980–2019. Combined with ERA5 reanalysis precipitation data, we quantified the AR-related precipitation over the region. We found that ARs are present during up to 5 % of the 6-hourly time steps in the most active areas. During these AR events, the region receives up to 40 % of the total annual precipitation. Additionally, the precipitation histograms show that the probability density is greater for the highest precipitation rates during AR events compared to non-AR events. By clustering the AR pathways using a k-means algorithm, we identified four typical AR pathways over Scandinavia (maximum annual AR frequencies and AR-related precipitation fraction in parentheses): over southern Denmark (4 %, 18 %), along the northern coast of Norway (2.5 %, 12 %), over the southern parts of Norway and the south-central parts of Sweden (1.8 %, 15 %), and along the southern coast of Norway (1 %, 7 %). Furthermore, we found that ARs over Scandinavia are typically most common during autumn and least frequent in spring, with some differences in seasonality between AR clusters. To investigate how large-scale atmospheric circulation affects Scandinavian ARs, we used the North Atlantic Oscillation (NAO) index to characterize circulation patterns during AR events. We found that AR activity over Scandinavia generally peaks during strong positive phases (>0.5) of the NAO. Our results indicate that ARs over Scandinavia, despite being relatively infrequent, are associated with a large fraction of the annual precipitation, which emphasizes their important role in the regional weather and climate.

## 1 Introduction

Atmospheric rivers (ARs) are long, but narrow, temporary pathways of unusually high atmospheric moisture transport that significantly influence precipitation patterns in many mid-latitude regions (e.g., Ralph and Dettinger, 2011). Globally, ARs are estimated to account for a majority of the global poleward moisture transport (Zhu and Newell, 1998), and have an important role in the local climate of many regions (e.g., Guan and Waliser, 2019). However, despite their potential importance for the Scandinavian climate system, ARs in the region have received limited scientific attention.



In ARs, moisture is transported long distances from mid-latitude ocean basins, in over land, and towards high latitudes (Zhu and Newell, 1998; Ralph et al., 2004). This long-range transport is a result of interactions between dynamical processes within the warm conveyor belt of extratropical cyclones (Gimeno et al., 2014). Here, in the pre-cold-frontal zone, the strong temperature gradient across the front results in a strong low-level jet, which combined with local moisture convergence and poleward moisture transport produces the intense moisture transport found within ARs (e.g., Ralph et al., 2005; Bao et al.,
30    2006).

ARs are an established feature in the mid-latitude climate. Along the west coast of North America, ARs are present for up to 13 % of the time annually, with slightly higher frequency during winter (November–March) (Rutz et al., 2014; Guan and Waliser, 2019). These ARs are associated with up to 50 % of the coastal precipitation (e.g., Ralph et al., 2004, 2005; Rutz and Steenburgh, 2012; Dettinger et al., 2011; Guan and Waliser, 2015), and have been linked to ending many persistent droughts
in the region (Dettinger, 2013).

Liang and Yong (2021) found that in South, Southeast, and East Asia there are up to 32 AR days per year ($\sim$8 % of the time) on average. In South and East Asia, ARs are most frequent during summer months, whereas autumn and winter constitute the most active AR season in Southeast Asia. Furthermore, they found that the AR contribution to precipitation varied with region and throughout the year. Over the coastal regions of East Asia, ARs are related to up to 32 % of the annul precipitation, while
ARs contribute between 24–36 % to the annual precipitation in South Asia. ARs over Asia have also been linked to more than 50 % of the most extreme precipitation days during the year (Liang and Yong, 2021; Kim et al., 2021).

In Western Europe, the annual AR frequencies are of a similar magnitude to those in North America. Here, maximum frequencies reach 13 % over the British Isles and parts of Norway, Spain, and Portugal (Guan and Waliser, 2019). Along these more coastal areas, the AR-related precipitation contributes to up to 30 % of the annual total precipitation, mostly during
autumn and winter (Lavers and Villarini, 2015). ARs have also been linked to many of the strongest annual precipitation events in Europe (Lavers and Villarini, 2013). Additionally, ARs have been linked to heavy winter floods in Britain (Lavers et al., 2012) and some of the largest floods in the lower Rhine basin (Ionita et al., 2020).

In high latitude regions, the effects of the moisture transported through ARs ranges from increased precipitation (Wille et al., 2021; Lauer et al., 2023) and snowfall (Gorodetskaya et al., 2014), to anomalously high surface ice melt over Greenland
(Mattingly et al., 2018) and Antarctica (Gorodetskaya et al., 2023). With the projected increases in temperature during the 21st century, AR frequencies are expected to increase at many of the previously mentioned locations (e.g., Ramos et al., 2016; Zhang et al., 2024; Shields et al., 2023; Hagos et al., 2016; Lavers et al., 2013), and as a result increase the importance of ARs in the climate of high-latitude regions. Increasing our understanding of the patterns and future changes of ARs is vital to improve the prediction of regional hydrological extremes.

Studies like the ones described above, investigating ARs on a regional to global level, commonly rely on some type of computerized detection of ARs in gridded datasets, such as reanalysis products or the output from climate models. Here, ARs are typically identified using a variable that describes the total amount of water vapour that is transported in the atmospheric column, such as the vertically integrated water vapour transport (IVT), in combination with an AR detection and tracking algorithm (ARDT). These algorithms typically operate by applying a threshold on the IVT field, either an absolute value or a





relative value derived from, e.g., a climatological quantile, to generate the initial selection of spatially continuous features of high IVT values. ARs are then identified by applying additional filtering criteria to these thresholded IVT regions, including constraints on geometric properties such as the length and width, average IVT direction, and temporal persistence. However, this relatively open choice when it comes to threshold and the properties used in the filtering has resulted in a large variety of ARDTs from the scientific community. To quantify the differences among the ARDTs, the Atmospheric River Tracking

Method Intercomparison project (ARTMIP; Shields et al., 2018) provides a common set of experiments on which the ARDTs can be evaluated and subsequently compared. One of the initial findings of the ARTMIP project is that there is a large spread in the AR detection rate among the participating ARDTs (Rutz et al., 2019). It is therefore recommended that studies should make use of multiple ARDTs whenever possible (Shields et al., 2023).

Previous studies focusing on ARs over Scandinavia found that up to 8 of the 10 strongest precipitation events along the coast

of Norway were related to ARs (Lavers and Villarini, 2013), and that moisture sources at low latitudes played an important role during extreme precipitation events (Stohl et al., 2008). However, a detailed climatology of AR activity over Scandinavia outlining the seasonality and local trends of ARs, as well as their spatial variations, is currently not available. Similarly, a descriptive view of the AR–precipitation relationship in the region is lacking.

Hence, the objectives of this study are to:

1. Quantify the contribution of ARs to regional precipitation.

2. Identify the different pathways of ARs that make landfall in Scandinavia.

3. Characterize the seasonal variability of AR activity.

4. Examine the relationship between large-scale circulation patterns and AR occurrence.

To accomplish this, we analysed all the ARs between 1980 and 2019 identified by four different ARDTs in ARTMIP. Section

2 describes the methodology and data used in this study. Section 3 presents the annual AR frequencies over Scandinavia, along with estimates of AR-related precipitation. In section 4, we examine the frequencies of the four identified AR pathways over Scandinavia and their influence on regional precipitation. Sections 5 and 6 then analyse the seasonality of these pathways and their relationship with large-scale circulation patterns, respectively. Section 7 discusses uncertainties and differences between ARDTs. Finally, section 8 summarizes the main findings and conclusions of the study.

## 2 Methods

### 2.1 Data

In this study, we used the results from four ARDTs (listed in Table 1) that are part of the ARTMIP project (Shields et al., 2018; Collow et al., 2022). These ARDTs were applied to the ERA5 reanalysis product (Hersbach et al., 2020) for the period 1980–2019 at hourly temporal resolution. For our analysis, we downsampled this to 6-hourly resolution to make the computations





feasible, which is in line with previous studies (e.g., Zhou et al., 2018; Guan and Waliser, 2017; Arabzadeh et al., 2020; Mattingly et al., 2018). We also used the 6-hourly accumulated total precipitation from the same ERA5 reanalysis product to analyse the influence of ARs on precipitation.

**Table 1.** ARDTs used in the study and their IVT thresholds.

| ARDT Name | IVT Threshold | Reference |
|---|---|---|
| GuanWaliser_v2 | IVT > 85th percentile | Guan and Waliser (2015, 2019) |
| Mundhenk_v3 | Anomalous IVT > 250 kg m$^{-1}$ s$^{-1}$ after removing mean and seasonal cycle | Mundhenk et al. (2016) |
| TempestLR | Laplacian of IVT $\geq$ 250 kg m$^{-1}$ s$^{-1}$ | McClenny et al. (2020); Rhoades et al. (2020); Ullrich and Zarzycki (2017) |
| Reid500 | IVT > 500 kg m$^{-1}$ s$^{-1}$ | Reid et al. (2020) |

ARDTs can be dived into two groups based on if they use an absolute IVT threshold or a relative IVT threshold. The absolute threshold uses a fixed value for the entire domain, e.g., IVT $\geq$ 500 kg m$^{-1}$ s$^{-1}$. For the ARDTs used in this study, only the Reid500 ARDT is classified as using an absolute threshold. The relative threshold ARDTs employ a threshold that can vary in space, and possibly time. For the remaining three ARDTs participating in this study, this is achieved in three different ways: GuanWaliser_v2 uses percentiles, Mundhenk_v3 uses anomalies above a fixed threshold, and TempestLR uses a fixed threshold applied to the Laplacian of the IVT field.

## 2.2 Identifying Scandinavian ARs

The four ARDT datasets share a common grid and temporal resolution, and were subsequently processed in the same way. In the following sections, this workflow will be described for a single ARDT dataset, but it was applied to all four ARDTs. We then aggregated the results of the four ARDTs, creating an ARDT ensemble, from which we derived the results of this study.

In the ARDT data, each time step contains a binary mask consisting of discrete areas, or "blobs", where ones represent identified ARs. To identify and track the individual ARs that make landfall in Scandinavia, we performed the following steps: First, we identified unique ARs in each time step using 4-connected component labelling to assign a unique identifier to each spatially coherent region of connected non-zero grid points. For this we used an implementation from the Python library Dask-Image. We will refer to each unique AR "blob" in a time step as an AR object. Second, we identified all AR objects that intersect Scandinavia at any point in time. Here, Scandinavia is defined as the geometric union of Denmark (excluding Greenland), Norway, and Sweden. To smooth out the irregular coastline of the region, the resulting shape was buffered two times: first with a distance of 0.85°latitude/longitude, and secondly with a distance of -0.15°(Fig. 1). Any AR object that at any point intersected this buffered version of Scandinavia was marked as making landfall. This buffering was done to capture ARs that do not directly intersect Scandinavia at the grid point level, but due to their proximity still likely have an effect on the precipitation over land. Here we limited the spatial domain to 50°to 74°N and 10°W to 45°E, as ARs extending beyond this region were deemed to have negligible impact on the characteristics of Scandinavian ARs. This also came with the added benefit





115 of lowering the computational requirements. Third, we tracked the AR objects that make landfall in Scandinavia backward in time, which is described next.

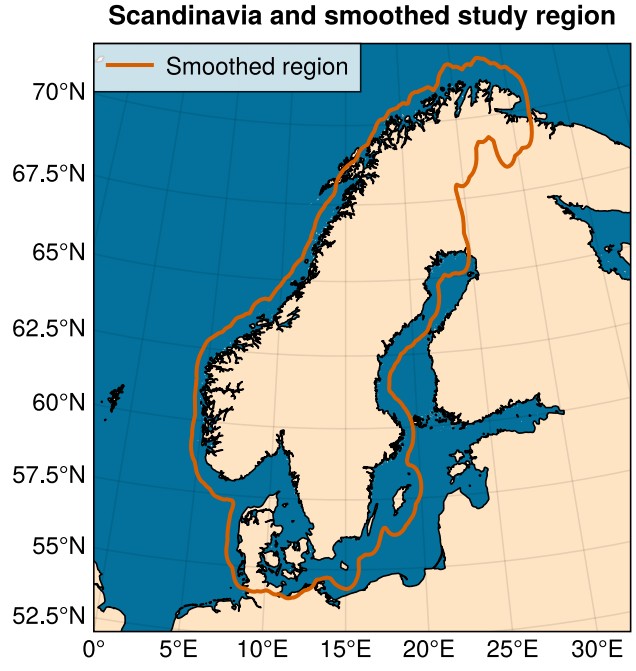

**Figure 1.** Map of the study region, here defined as the geometric union of Denmark, Norway, and Sweden. The orange line outlines the buffered region which was used to identify ARs that intersect Scandinavia.

The ARDT datasets do not contain information about the temporal relationships between AR objects across consecutive time steps. Thus, to track ARs over time, we developed a method to identify which AR objects in different time steps belong to the same AR event. Our tracking algorithm uses an iterative approach, similar to the one described in Guan and Waliser

120 (2019). It compares each AR object in the current time step to all AR objects in the previous time step and finds the most similar pairs. This pair is then considered as part of the same AR event. The similarity of the AR objects was computed using an area-weighted Jaccard index (as described in Deza and Deza, 2016), which measures the fraction between the intersection and union of two samples:

$$J = \frac{|A \cap B|}{|A \cup B|}, \tag{1}$$

125 where $A$ and $B$ are two AR objects from separate time steps. The horizontal bars denote the absolute value operator. The Jaccard index is 0 for two objects with no intersection, and 1 for objects that completely overlap. An AR object in the current time step is considered a possible continuation of any previous AR object that returns a positive Jaccard index. In the case when two AR objects have a non-zero Jaccard index with more than one AR object in the previous time step, the AR object with





the highest Jaccard index is considered its continuation. Similarly, if an AR object has a non-zero Jaccard index with multiple
AR objects in the previous time step, the AR object with the highest index is considered as the origin. Our approach differs
from that of Guan and Waliser (2019) in that we performed the tracking on the binary AR masks from ARTMIP rather than on
the IVT field. Furthermore, we also allowed the size of the look-back during tracking to iteratively increase from 6 up to 24
hours if no matching ARs were found. This was done to account for the possibility of AR objects missing in a time step due to
the strict geometry constraints of the ARDT. Hence, if two overlapping AR objects are separated by e.g. 12 hours with no AR
object in-between, they were still considered to be part of the same AR event.

Finally, we summed all temporally linked AR objects to create composite AR pathways representing the full lifecycle of
individual AR events. These AR pathways were then used to analyse the general AR characteristics over Scandinavia, such as
frequencies and trends, as well as in the identification of the AR clusters.

## 2.3 Clustering ARs

To investigate the possible recurring patterns of AR pathways over Scandinavia, we categorised the concatenated AR pathways
using k-means clustering, analysing each ARDT separately. The k-means algorithm requires the user to specify the numbers
of clusters that should be fit to the data. To find the optimum number of clusters we evaluated the sample silhouette scores
from two to ten clusters (see Figs. A11–A14). We found the optimal number of initial clusters to be different depending on the
ARDT: four for GuanWaliser_v2 and Mundhenk_v3, and six for Reid500 and TempestLR.

The clusters were initialized and fit using the "Scalable K-Means++" algorithm described in Bahmani et al. (2012) and im-
plemented in the Python library Dask-ML. After the initial fit, we observed that some of the clusters in Reid500 and TempestLR
exhibited high spatial correlation. To consolidate these clusters, we computed the spatial correlation between each cluster and
the four clusters of Mundhenk_v3, which served as a reference. This resulted in four clusters for each ARDT, which were
re-ordered based on their spatial correlation with the Mundhenk_v3 clusters. These clusters represent the common pathways
of ARs that make landfall in Scandinavia.

## 2.4 ARs and regional precipitation

To investigate the relationship between ARs and precipitation in Scandinavia, we identified all time steps during which ARs
make landfall in Scandinavia. Based on the AR time steps we calculated the AR-related precipitation by subtracting the average
non-AR precipitation from the precipitation during AR events, while also capping the AR precipitation $\geq 0$. We then computed
both total annual AR-related precipitation and average 6-hourly precipitation rates for all AR events and for each AR cluster.
To evaluate the spatial co-location between ARs and precipitation, we computed the field correlations between AR frequency
and precipitation patterns.




## 2.5 Grouping AR pathways by season and NAO index

To examine the variability of ARs and AR-related precipitation, we grouped the AR pathways by the assigned cluster and the season. Here, we used the standard meteorological season definitions: December–January–February (DJF), March–April–May (MAM), June–July–August (JJA), and September–October–November (SON). Each AR pathway was assigned a season based on the time when it first made landfall in Scandinavia.

To further investigate how the large-scale circulation affects ARs over Scandinavia, we used the North Atlantic Oscillation (NAO) index, as this is the leading mode of variability of the atmospheric circulation over the North Atlantic and Northern Europe (e.g., Hurrell and Deser, 2010). We used the normalized daily NAO index from the Climate Prediction Centre (NOAA Climate Precirion Center, 2025) to represent the phase of the NAO. The daily mean NAO index was assigned to each AR pathway based on the first time the AR made landfall in Scandinavia. To group the AR pathways by their respective NAO indices, we first divided the range of NAO values into four groups of roughly equal size, centred at 0. This resulted in four NAO bins containing days with either a: strong negative ($-3.5 < \text{NAO} \le -0.5$), negative ($-0.5 < \text{NAO} \le 0$), positive ($0 < \text{NAO} \le 0.5$), or strong positive ($0.5 < \text{NAO} \le 3.5$) phase of the NAO. We then grouped the AR pathways by their assigned NAO-bin and cluster. For each cluster and NAO-bin combination, we computed the average AR frequencies by counting the total number of AR time steps and dividing it by the total number of time steps in the corresponding NAO bin. To further quantify the spatial redistribution of AR frequencies for the different phases of the NAO, we computed the mean Wasserstein distance between the four frequency distributions. For each cluster, we calculated the distances from each NAO-bin to all other bins and computed the mean of the results to obtain a scalar measure of the total spatial distribution change.

## 3 Annual AR frequencies and AR-related precipitation over Scandinavia

The analysis outlined in the methods resulted in four separate sets of results, one for each ARDT, of annual AR frequencies, AR clusters, and AR-related precipitation estimates. We refer to these four results as the ARDT ensemble and present the ensemble median values, along with the range spanned by the lowest and highest member within brackets. In the median of this ARDT ensemble, ARs that make landfall in Scandinavia are present for up to 5 % of the time during an average year (Fig. 2a). At the same time, these ARs are associated with up to 40 % (1050 mm) of the annual average local precipitation (Fig. 2b & c) in the region. Specifically, the regions with the highest AR-related precipitation fraction are found along the southwest coast of Norway and the west coast of Denmark. Furthermore, we note that over Scandinavia in its entirety, the minimum AR-related precipitation fraction is 12 %. The AR frequency and precipitation patterns are moderately well aligned, with a median field correlation of 0.46 across all ARDTs.

Compared to the AR frequency maximum, the location of the maximum AR-related precipitation is shifted further north, toward the Norwegian southwest coast. The high AR activity over the west coast of Denmark does not enhance precipitation over Denmark to the same degree. This difference can likely be explained by orographic effects: Norway's mountainous terrain enhances AR precipitation through orographic lifting (e.g., Roe, 2005), while Denmark's relatively flat topography allows ARs to reach further inland (e.g., Rutz et al., 2014).





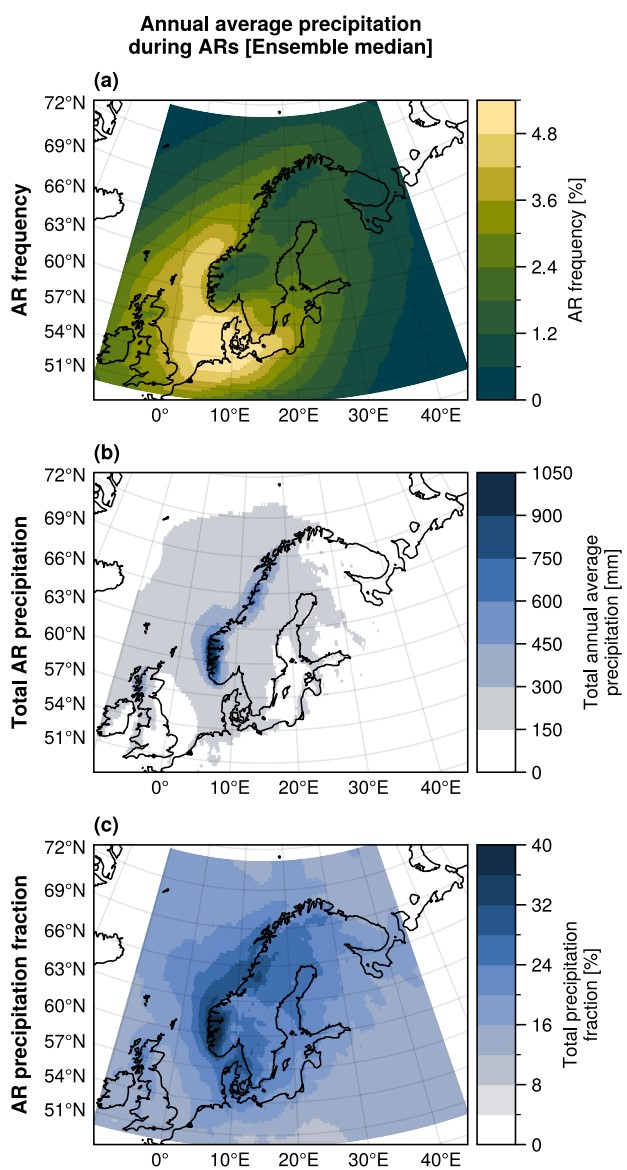

**Figure 2. (a)** Annual average AR frequency, **(b)** annual total AR-related precipitation, and **(c)** fraction of total precipitation attributed to ARs over Scandinavia between 1980 and 2019. All values represent the median of the ARDT ensemble, which comprises four AR detection algorithms: Mundhenk_v3, GuanWaliser_v2, Reid500, and TempestLR. Both the AR frequency and precipitation aggregates are based on ERA5 reanalysis data.

To further examine the relationship between the annual ARs and precipitation, we computed the annual frequencies of ARs making landfall anywhere over Scandinavia and compared these with the spatial average of annual precipitation and AR-related precipitation over the same region (Fig. 3). The ARDT ensemble median shows that ARs intersect Scandinavia between 20 %



and 30 % of the time during the year. For the ensemble maximum, these numbers increase up to between 35 % and 55 %, while
for the ensemble minimum it remains below 10 %. It is most common that a single AR intersected Scandinavia at any given
time, with multiple ARs intersecting the region simultaneously during only 2 % of time steps on average. These relatively high
AR frequencies compared with the lower spatial AR frequencies in Fig. 2 indicate that ARs intersect Scandinavia via different
pathways.

The total annual AR frequency is moderately correlated with the annual total precipitation (light blue bars in Fig. 3) over
Scandinavia, with an ensemble median [min, max] $r$-value of 0.46 [0.35, 0.48], all significant at $p < 0.05$. Unsurprisingly, the
total AR frequency is strongly correlated to the annual total AR-related precipitation, with an $r$-value of 0.92 [0.88, 0.98]. Fur-
thermore, the correlation between the regional annual total precipitation and AR-related precipitation is high, with a significant
($p < 0.05$) $r$-value of 0.68 [0.40, 0.77].

We investigated the trends in the annual AR occurrence using the Mann-Kendall trend test, and did not find any significant
trends for any of the ARDTs.

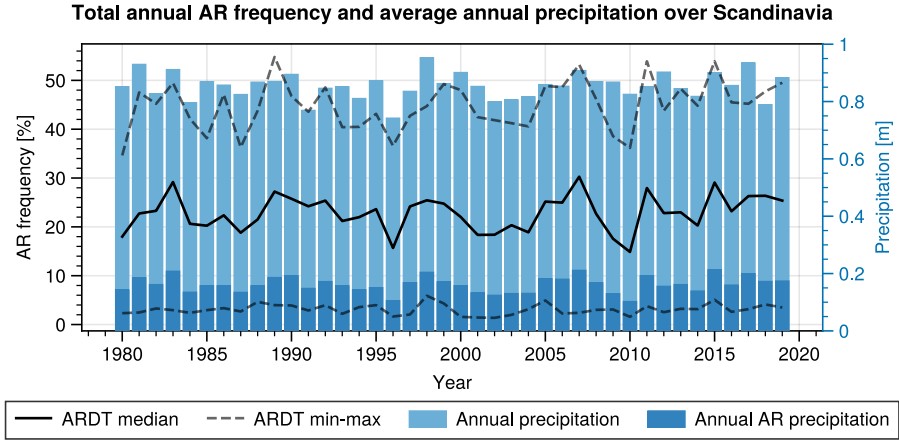

**Figure 3.** Time series of total annual AR occurrence and annual precipitation over Scandinavia. The black solid line shows the ARDT
ensemble median AR occurrence, while the dashed lines show the ARDT ensemble minimum and maximum. The bars, which both begin at
0, show the spatial averages of annual total precipitation (light blue bars) and AR-related annual total precipitation (blue bars).

Comparing the 6-hourly accumulated precipitation during AR and non-AR events, we find that high precipitation amounts
(more than 12 mm 6h$^{-1}$) are more common during AR events compared to non-AR events (Fig. 4a & b). Specifically, for the
spatially averaged precipitation we find that precipitation rates are generally lower during ARs in the ARDT ensemble median.
If we instead look at the ARDT ensemble maximum, precipitation rates $\geq 0.66$ mm 6h$^{-1}$ are more common during ARs. We
also examined the spatial maximum of precipitation to understand how ARs influence the more intense precipitation over the
region (Fig. 4b). Our results show that precipitation $\geq 12$ mm 6h$^{-1}$ is more common during AR events, even in the ARDT
ensemble minimum.



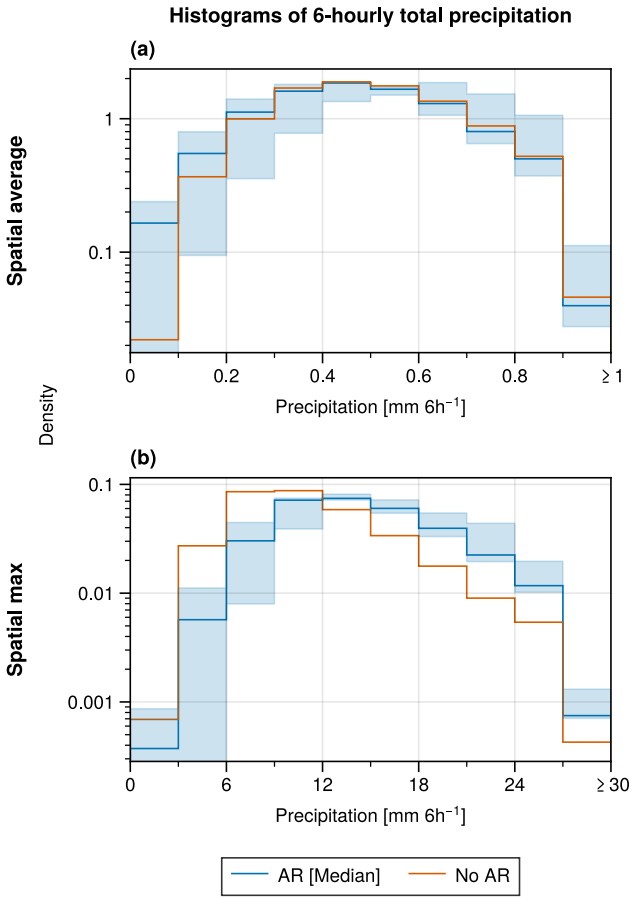

**Figure 4.** Distributions of the spatial **(a)** average and **(b)** maximum 6-hourly accumulated precipitation during AR (blue) and non-AR (orange) events. The rightmost bins in the histograms are open-ended, encompassing all values $\geq 0.9$ mm 6h$^{-1}$ in **(a)** and $\geq 27$ mm 6h$^{-1}$ in **(b)**. Shading indicates the full range of the ensemble for the AR conditions.

## 4 Primary AR pathways over Scandinavia

We identified four AR clusters that represent common pathways of ARs that make landfall in Scandinavia. Similarly to the

preceding analysis, the clustering was done on each ARDT separately, and then the results were combined into an ensemble. The ensemble median annual average AR frequency and AR-related precipitation for each cluster are shown in Fig. 5.

The southernmost ARs are found in cluster 1 (Figs. 5a & b, A3a & b, A4a & b). This is the most common pathway for ARs reaching Scandinavia, with annual frequencies reaching up to 4 % [0.8–5 %], or ∼ 15 days. Here, the AR frequency maximum is located mostly outside Scandinavia, in the region between Denmark and Northern Germany. Cluster 1 is also where we

find the highest AR-related precipitation (18 % [6.4–25 %]). Compared to the AR frequency pattern, the pattern of AR-related



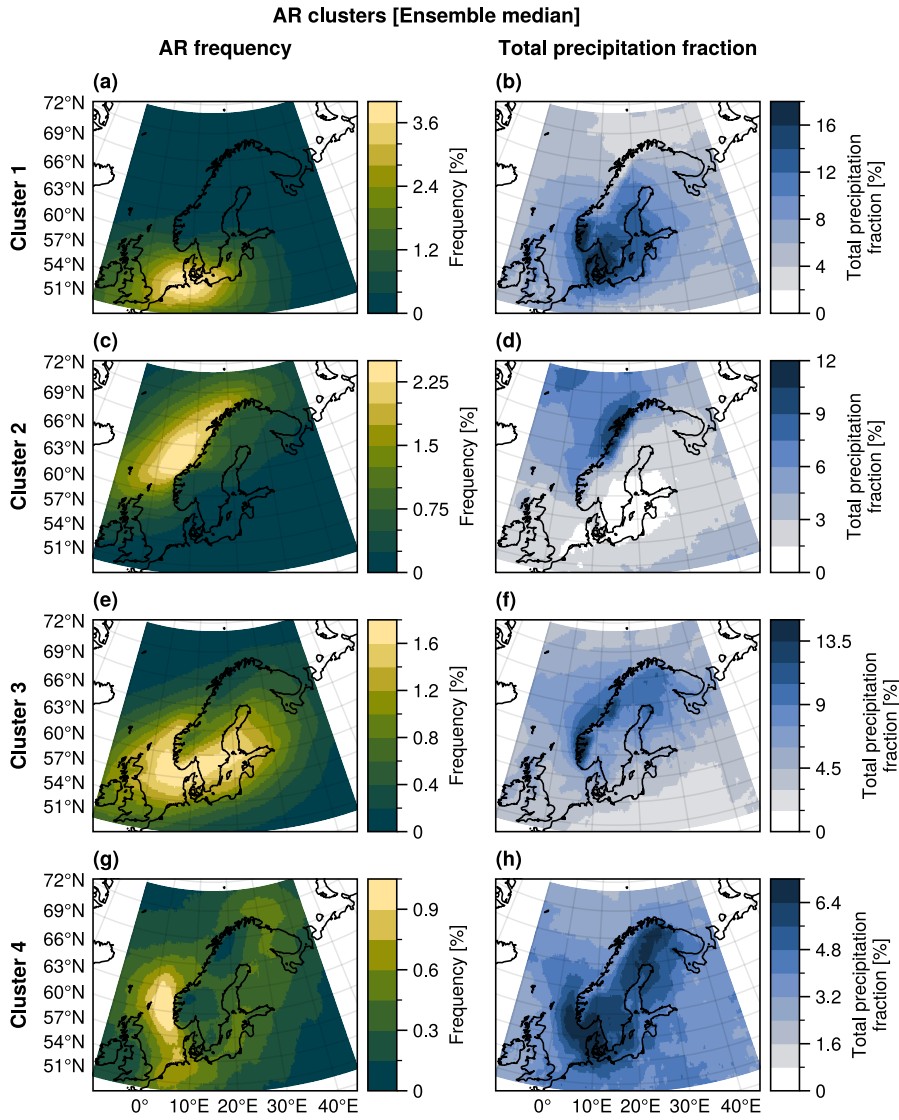

**Figure 5.** Ensemble median annual average AR frequencies (left column) and total precipitation fraction (right column) for each identified AR cluster (rows). Note that the colour scales differ between the subplots.

precipitation is shifted slightly to the north, over Denmark and southern Sweden, but the field correlation between the patterns remains moderately high: 0.52 [0.34–0.67].

Cluster 2 (Figs. 5c & d, A3c & d, A4c & d) captures ARs that make landfall along the northern west coast of Norway. These ARs occur up to 2.5 % [0.4–4.0 %] of the time during an average year ($\sim$ 9 days) and are related to a maximum of 12 % [3–15 %] of the annual precipitation in the area. This cluster shows the highest field correlation between the AR frequency and precipitation patterns: 0.62 [0.50–0.75].






In cluster 3, ARs are located along the southern west coasts of Norway, Sweden, and Denmark, with ARs also extending further inland over Sweden (Fig. 5e & f). During the average year, these ARs occur up to 1.8 % [0.4–6.0 %] of the time (∼ 6 days, Figs. 5e, A3e, A4e). Here, the AR-related precipitation accounts for up to 15 % [3–25 %] of the total annual precipitation.

The area of high AR-related precipitation corresponding to cluster 3 is shifted slightly to the north compared to the AR frequency pattern, resulting in a relatively low field correlation of 0.37 [0.06–0.76] between AR frequency and precipitation patterns. The ARs in cluster 3 are also a good example of the topographical constraints imposed by the Norwegian mountain range on AR propagation; the ARs appear to be blocked from propagating over the Norwegian mountains, and instead traverse the North Sea and the Skagerrak Strait toward the Swedish west coast, where the flat topography enables the ARs to penetrate

further inland and even reach the Baltic Sea and beyond.

Cluster 4 (Figs. 5g, A3g, A4g) captures ARs that mainly occur west of Denmark's west coast and Norway's southwest coast, with a predominant south-to-north orientation. These pathways exhibit some overlap with cluster 3, but the ARs in cluster 3 have a more west-east orientation. The ARs in cluster 4 are relatively infrequent, and are only present for up to 1.0 % [0.2–1.4 %] of the time (or ∼ 3 full days in a year). Despite this, the ARs in cluster 4 are related to a relatively large fraction of the total

annual precipitation in the area of high AR frequency: up to 7.3 % [1.4–10.0 %] (Figs. 5h, A3h, A4h). Furthermore, for cluster 4 we find that the field correlation between the AR frequency and precipitation patterns is relatively high: 0.48 [0.44–0.56].

## 5   Seasonality of AR pathways over Scandinavia

The AR frequencies, and to some extent location, of the identified AR clusters vary throughout the year. In the following section we present the seasonal AR frequencies for each cluster. The frequencies represent the fraction between the number of

AR time steps during each season and the total number of time steps in the respective season.

For all clusters, ARs are the least frequent during the spring (MAM, Fig. 6b,f,j,n), while autumn (SON) is overall the most active AR season (Fig. 6d,h,l,p). In cluster 1, AR activity is high throughout most of the year, with AR frequencies exceeding 3.0 % [0.6–4.2 %] (Figs. 6a–d, A5a–d, A6a–d) during all seasons. However, these southernmost ARs still exhibit a weak seasonal cycle, characterized by a dip in the AR activity during spring (MAM), followed by a gradual increase in AR

activity towards autumn (SON), where AR frequencies reach 5.0 % [1.6–6.0 %]. We also find this pattern in the monthly total AR frequency for cluster 1, where most ARDTs shows a valley during April and May (Fig. 7a). In the ensemble maximum, however, the high AR frequency areas during summer and autumn are located further towards the east over the Baltic Sea (Figs. 6c & d, A6c & d).

Cluster 2 shows a minimum in AR activity during spring (MAM), where AR frequencies reach 2.0 % [0.5–4.5 %] over

a small area (Fig. 6f). During winter (DJF), the AR frequencies in this region are slightly higher and reach 2.4 % [1.1–4.5 %]. However, the area of high frequencies is small compared to the areas of high frequencies during both summer (JJA) and autumn (SON). For these two seasons, AR frequencies reach 3.2 % [1.2-4.5 %] over a long stretch of the Norwegian coastline (Figs. 6g & h, A5g & h, A6g & h). This seasonality is also visible in the monthly total AR frequency for cluster 2, which for two ARDTs (Mundhenk_v3 & TempestLR) show an increase in AR activity during the summer months, followed by a slow





decrease towards the minimum in April and May (Fig. 7b). This seasonal cycle is not as noticeable in GuanWaliser_v2, which

exhibits a period of slightly increasing activity during summer, followed by a series of valleys and peaks during the rest of the

year.

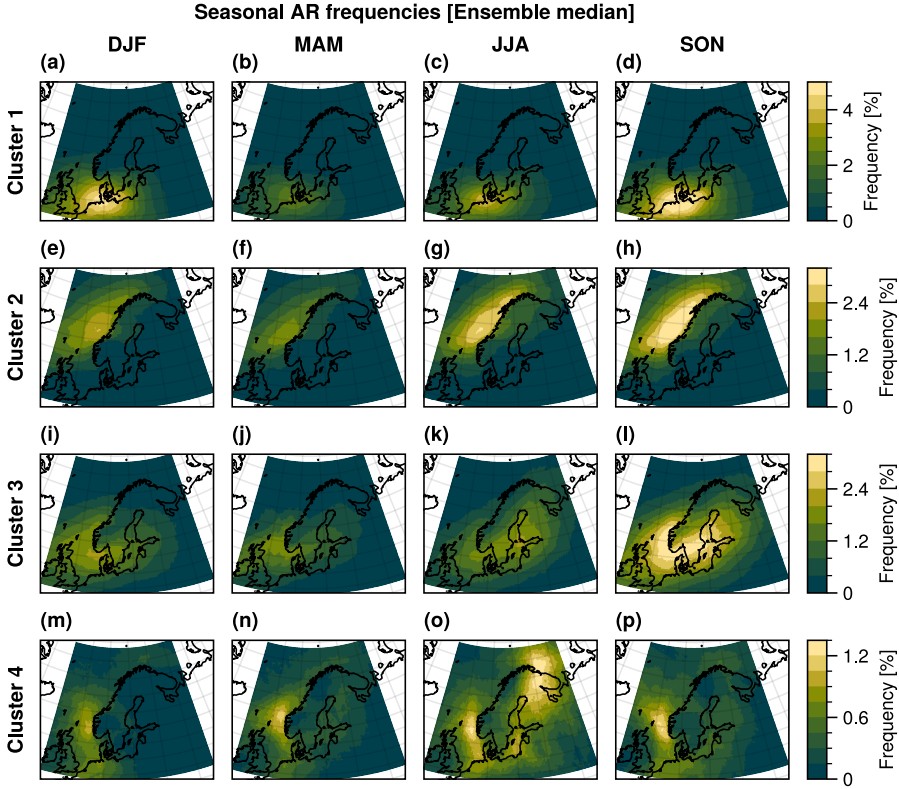

**Figure 6.** Maps of the average annual AR frequencies (ensemble median) during the seasons: winter (DJF), spring (MAM), summer (JJA), and autumn (SON) for each AR cluster. The frequencies correspond to the fraction between the number of AR time steps in each season and the total number of time steps in each respective season.

Similar to cluster 2, AR activity in cluster 3 peaks during the autumn. Here the area of high AR frequencies extends from the British Isles, across Sweden, and towards Finland, with AR frequencies reaching 3.2 % [1.2–8.0 %] (Figs. 6i–l, A5i–

l, A6i–l). Following autumn, ARs are less frequent during winter and summer, with AR activity reaching a minium during spring. Comparing the AR activity during winter and summer, the area of high AR frequency is shifted from the North Sea and Skagerrak Strait further east over the mainland of Sweden and the Baltic Sea (Fig. 6i & k). For cluster 3, the monthly total AR frequencies show a distinct peak during September and October in all ARDTs except Reid500 (Fig. 7c).

The ensemble median AR activity in cluster 4 is relatively low throughout the year but shows a slight increase during

summer (JJA) and autumn (SON), when frequencies reach ∼1.4 % [0.3–2.0 %] (Figs. 6m–p, A5m–p, A6m–p). Similarly,





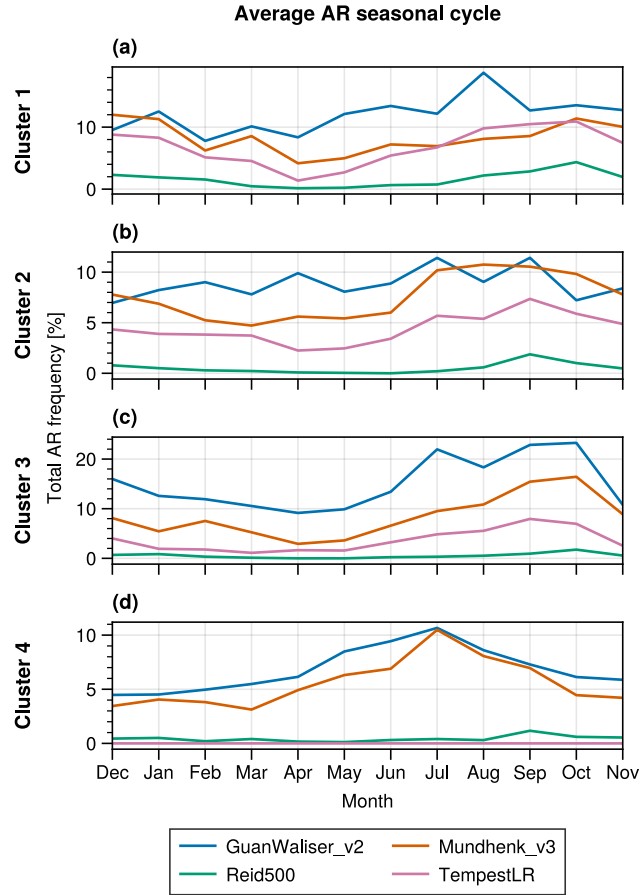

**Figure 7.** Average monthly total AR probability for the four AR clusters (rows). Each line represents a single ARDT: GuanWaliser_v2 (blue), Mundhenk_v3 (orange), Reid500 (green), and TempestLR (pink).

the monthly total AR frequency show a clear seasonal cycle for two ARDTs (GuanWaliser_v2 and Mundhenk_v3), with low activity throughout most of the year but with a clear peak in July (Fig. 7d).

## 6    Influence of NAO on AR pathways

Finally, we examined the relationship between NAO and the frequency of AR pathways. The AR frequencies for each NAO-bin
and AR cluster combination represent the relative AR occurrence the respective NAO phase. Overall, the four clusters show the highest AR activity during strong positive phases of the NAO ($> 0.5$; Fig. 8d,h,l,p). However, the magnitude of AR activity during the weaker phases of the NAO varies among the clusters. For cluster 1, we see relatively high AR frequencies for all phases of the NAO, with the highest frequencies during the strongly positive phase. These ARs are the least frequent during



the weakly negative phase of the NAO, where frequencies reach 2.4 % [0.4–3.2 %] (Figs. 8b, A7b, A8b). Interestingly, ARs in
cluster 1 appear more frequently during the strongly negative phase of the NAO compared to the weakly positive and negative
phases, especially in the ARDT ensemble maximum (Fig. A8a–c). The NAO-grouped AR distributions for cluster 1 show
moderate changes between the groups, as seen by the mean Wasserstein distance (Fig. 9).

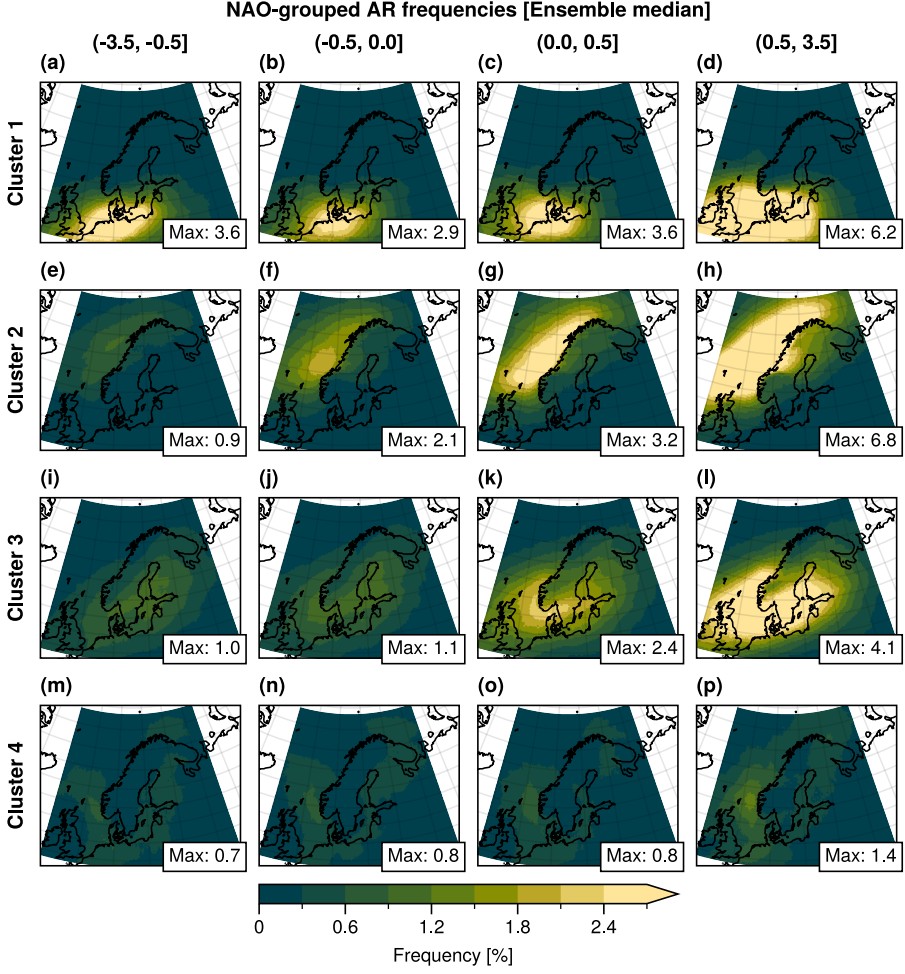

**Figure 8.** Maps of the average AR frequency (ensemble median) during the different phases of the North Atlantic Oscillation (NAO) for each
AR cluster. The NAO phases are categorized into four groups: strongly negative (first column), weakly negative (second column), weakly
positive (third column), and strongly positive (fourth column). The corresponding NAO index intervals are shown above each column. The
colour scale is capped at 2.7 %; the maximum frequency value for each subplot is displayed in the bottom right corner.

A majority of the AR activity in cluster 2 takes place during the positive phases of the NAO, where AR frequencies exceed
2.4 % [0.4–8.0 %] (Figs. 8e–h, A7e–h, A8e–h). Here we note that the maximum AR frequencies more than doubles between
the positive and strongly positive phases. The frequency difference between the positive and strongly positive phase is also





slightly strengthened in the ARDT ensemble maximum (Fig. A8g & h), where the maximum frequency difference is 4.3 percentage points. Cluster 2 also shows the highest mean Wasserstein distance, which indicates that the AR frequencies in cluster 2 changes the most with NAO (Fig. 9a–b).

Similar to cluster 2, ARs in cluster 3 are more frequent during the positive phases of the NAO. Here, AR frequencies reach
2.4 % [0.4–8.0 %] during the positive phase of the NAO, while surpassing 2.4 % [0.4–8.0 %] during the strong positive phases (Figs. 8k & l, A7k & l, A8k & l). For the two bins with ARs during negative phases of the NAO, frequencies reach 1.2 % [0.1–3.2 %] (Figs. 8i & j, A7i & j, A8i & j) over a limited area. For cluster 3 we note that there is a relatively large difference in both the shape and location of the AR frequency pattern between NAO bins. The mean Wasserstein distance for cluster 3 is high, similar to cluster 2 (Fig. 9). Interestingly, in the ensemble maximum (Fig. A10), the mean distances surpasses those of
cluster 2.

In cluster 4, during the weaker phases of the NAO, differences in AR activity are relatively small, with frequencies reaching ∼1.2 % [0.2–1.6 %] (Figs. 8m & n, A7m & n, A8m & n). During the strong positive phase of the NAO, AR frequencies reaches up to 1.8 % [0.3–2.4 %] (Figs. 8p, A7p, A8p). These relatively small changes in AR activity with the NAO are also reflected in the mean Wasserstein distance for cluster 4 (Fig. 9).

Overall, comparing the southern ARs in cluster 1 to the more northern-reaching ARs, most notably cluster 2, the northern ARs appear to be more strongly linked to the NAO. This is consistent with previous research, which found that ARs in Northern Europe are more frequent during the positive phase of the NAO (Lavers and Villarini, 2013). A possible explanation for this is that the more southern ARs are primarily governed by the climatological mean flow, whereas the more northerly ARs that make landfall in clusters 2 and 3 are strongly influenced by the enhanced westerlies during positive phases of the NAO.

Regarding the seasonality of the ARs found in cluster 2, compared to the seasonality of the Northern Hemisphere storm tracks, where intensity peaks during autumn and winter (e.g., Hoskins and Hodges, 2019), the peak season of these north-ernmost ARs appears to be shifted slightly more towards the summer months. This suggests a possible different mechanism driving the seasonality of these northern ARs, potentially the higher water content of the warmer summer atmosphere. As we have shown, AR activity is also strongly driven by the NAO, but the seasonal cycle of the NAO does not coincide with the
observed seasonal cycle of ARs.

## 7 ARDT ensemble spread and uncertainties

The spread in the detected AR frequency within the ARDT ensemble is relatively large. The difference between the members with highest and lowest annual AR frequencies is 9 percentage points (Figs. A1a & A2a). Similarly, the ensemble range for the AR-related precipitation fraction is 44 percentage points, or 1200 mm. The large spread is also evident in the field correlation
between AR frequency and AR-related precipitation patterns, which ranges from 0.09 and 0.78. This large difference in the AR detection rate, and subsequent AR-related precipitation, among the ARDTs is largely determined by the IVT threshold employed by the ARDT (Rutz et al., 2019). Here we note that the GuanWaliser_v2, Mundhenk_v3, and TempestLR ARDTs employ thresholds that vary in space, allowing them to detect ARs at higher latitudes, where atmospheric water vapour and





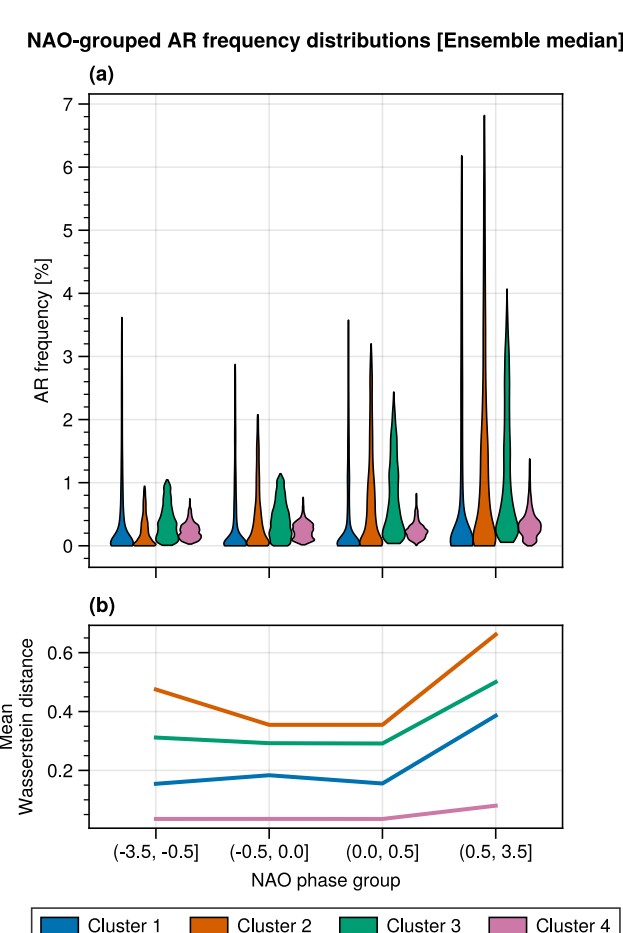

**Figure 9.** AR frequency distributions grouped by **(a)** NAO phase and **(b)** the mean Wasserstein distance between the successive distributions for each AR cluster (ensemble median). The distributions in **(a)** are visualized using violin plots. The height of the violin covers the whole range of values in the distribution, while the width at any given value indicates the relative frequency of that value in the distribution.

IVT are typically lower. In contrast, the Reid500 ARDT employs an absolute IVT threshold, which limits AR detection in these regions. Overall, the percentile-based GuanWaliser_v2 algorithm produces the highest AR frequencies, whereas the Mundhenk_v3 and TempestLR ARDTs yield relatively similar results. Interestingly, Mundhenk_v3 and TempestLR show a similar seasonal cycle for the southernmost cluster (Figs. 6a, 7a), while the differences are larger in the more northern clusters (Figs. 6b & c, 7b & c).

In the relatively high latitude region around Scandinavia, overall atmospheric conditions are drier compared to lower latitudes. Considering these conditions, ARDTs using a relative threshold will be more suitable to identify ARs in this region, since they account for the background moisture content. There are still differences among the relative threshold ARDTs, but overall they agree on the location and seasonal cycle of ARs that intersect Scandinavia.



In this study we have used four different ARDTs applied on a single reanalysis dataset (ERA5). Using multiple reanalysis products would increase the robustness of our results by enabling us to explore the uncertainties introduced by the choice of reanalysis product. However, while there are notable differences in for instance the mean IVT in different reanalysis products, the differences in global AR detection due differences among ARDTs are overall larger (Collow et al., 2022). Therefore, we considered the use of a single reanalysis dataset to be sufficient for our analysis.

This study relies on tracked ARs in order to improve the clustering of ARs. Notably, the ARTMIP catalogue does not provide ARDT native tracking data for the ARs. Instead, we have performed our own tracking of the AR objects. It is important to note that we have not implemented the tracking methods specific to each ARDT; instead, we have used an implementation based on the tracking method of the GuanWaliser_v2 ARDT. It is possible that this has introduced some differences compared to the original tracking. However, it was not considered feasible to re-implement four different tracking algorithms for this study. We suggest that incorporating tracking information into the ARTMIP catalogue would be a valuable enhancement.

## 8 Conclusions

In this study we have presented a climatology of ARs that make landfall in Scandinavia during 1980–2019. The results are based on the output from four different AR detection algorithms from the ARTMIP project, which were applied to ERA5 reanalysis data. The values presented in this section represent the median from analyses conducted with these four ARDTs. In connection to generating the AR climatology, we have used ERA5 precipitation to evaluate how Scandinavian ARs affect the regional precipitation.

We found that ARs are a common feature of the Scandinavian climate, with a considerable impact on the regional precipitation. Looking at the entire region, AR activity peaks over Denmark, where the average annual AR frequencies reach 5 % of the time, corresponding to around 18 AR days per year. These areas of relatively high AR frequencies extend both west towards the southwest coast of Sweden, and north towards the southwest coast of Norway. These areas show only slightly lower annual AR frequencies between 3.6 % to 4.8 %. For the entirety of Scandinavia, ARs that intersect the region are present for 25 % of the time during the average year.

The corresponding spatial average AR-related precipitation is 200 mm, or 20 % of the annual total. Notably, the AR-related precipitation is focused along the mountainous coast of Norway, most likely due to orographic uplift. However, even far inland across Norway, Sweden, and Denmark, AR-related precipitation still constitutes a substantial fraction (>24 %) of the annual total precipitation.

We identified four frequent pathways of ARs that reach and intersect Scandinavia: (1) ARs from southwest reaching southern Denmark; (2) ARs from the southwest reaching the northern west coast of Norway; (3) ARs from the west reaching southern Norway and the west coast, and further inland across to the east coast, of Sweden; and (4) ARs from the south reaching the southwest coast of Norway. Out of these four pathways, the ARs reaching southern Denmark are the most frequent, followed by the ARs reaching northern Norway. The spatial maximum AR-related precipitation values for the four pathways correspond to 18 %, 12 %, 15 %, and 7 % of total annual precipitation, respectively. All four common AR pathways share a similar seasonal




cycle, with AR activity generally peaking during autumn and reaching a minimum during spring, with no significant long-term trends during the study period.

Scandinavian ARs show a strong positive connection to the North Atlantic Oscillation. For all four AR pathways, the AR activity peaks during the strong positive NAO phases (NAO index > 0.5). However, we also observe spatial variation in this relationship: the northernmost ARs exhibit a more linear relationship with NAO index, while the southern ARs show weaker sensitivity during negative and weakly positive NAO phases.

Our analysis reveals that there are large differences in the results from the four ARDTs. For the annual AR frequency, the difference between the ARDTs with the lowest and highest AR activity is around 9 percentage points, while the maximum AR-related precipitation fraction varies between 10 % and 54 %. These differences are largely explained by whether the ARDT employs a fixed or a relative IVT threshold, with ARDTs using relative thresholds yielding higher AR activity. However, there are also noticeable differences within this group, where the annual AR frequency varies by around 5 percentage points. This large spread underscores the importance of evaluating multiple detection algorithms in AR studies.

The significant role of ARs on precipitation over Scandinavian, as shown here, raises the question of whether diagnosing and tracking ARs in numerical weather prediction could improve the accuracy of local precipitation forecasts. Although such approaches have been evaluated for other regions in the world (e.g., Wick et al., 2013; Nayak et al., 2014), dedicated implementation efforts would be needed for Scandinavia. Additionally, investigating how these Scandinavian ARs and their regional climate impacts will change throughout the 21st century remains an important area for future research.

*Code availability.* Code used in the analysis for this paper is available on GitHub: https://github.com/Holmgren825/ARs_Scandinavia

*Data availability.* ARTMIP datasets are available for download from the Research Data Archive (NSF NCAR). ERA5 reanalysis data is available for download from the Copernicus Climate Data Store (Copernicus Climate Change Service).





## Appendix A: Figures

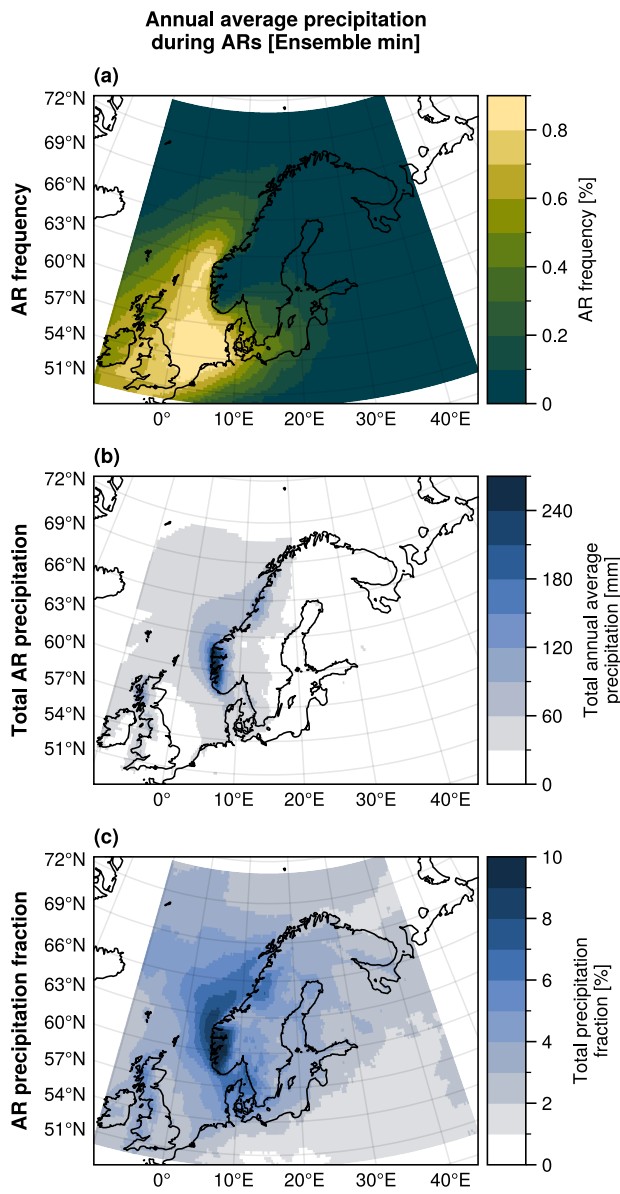

**Figure A1.** **(a)** Annual average AR frequency, **(b)** annual total AR-related precipitation, and **(c)** fraction of total precipitation attributed to ARs over Scandinavia between 1980 and 2019. All values represent the minimum of the ARDT ensemble, which comprises four AR detection algorithms: Mundhenk_v3, GuanWaliser_v2, Reid500, and TempestLR. Both the AR frequency and precipitation aggregates are based on ERA5 reanalysis data.

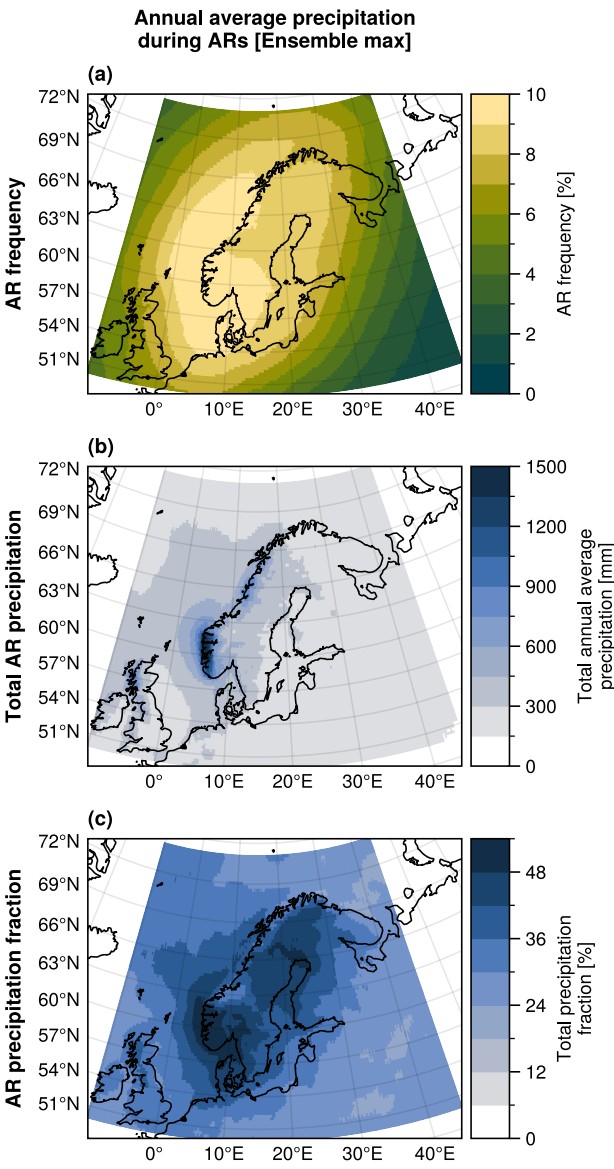

**Figure A2. (a)** Annual average AR frequency, **(b)** annual total AR-related precipitation, and **(c)** fraction of total precipitation attributed to ARs over Scandinavia between 1980 and 2019. All values represent the maximum of the ARDT ensemble, which comprises four AR detection algorithms: Mundhenk_v3, GuanWaliser_v2, Reid500, and TempestLR. Both the AR frequency and precipitation aggregates are based on ERA5 reanalysis data.

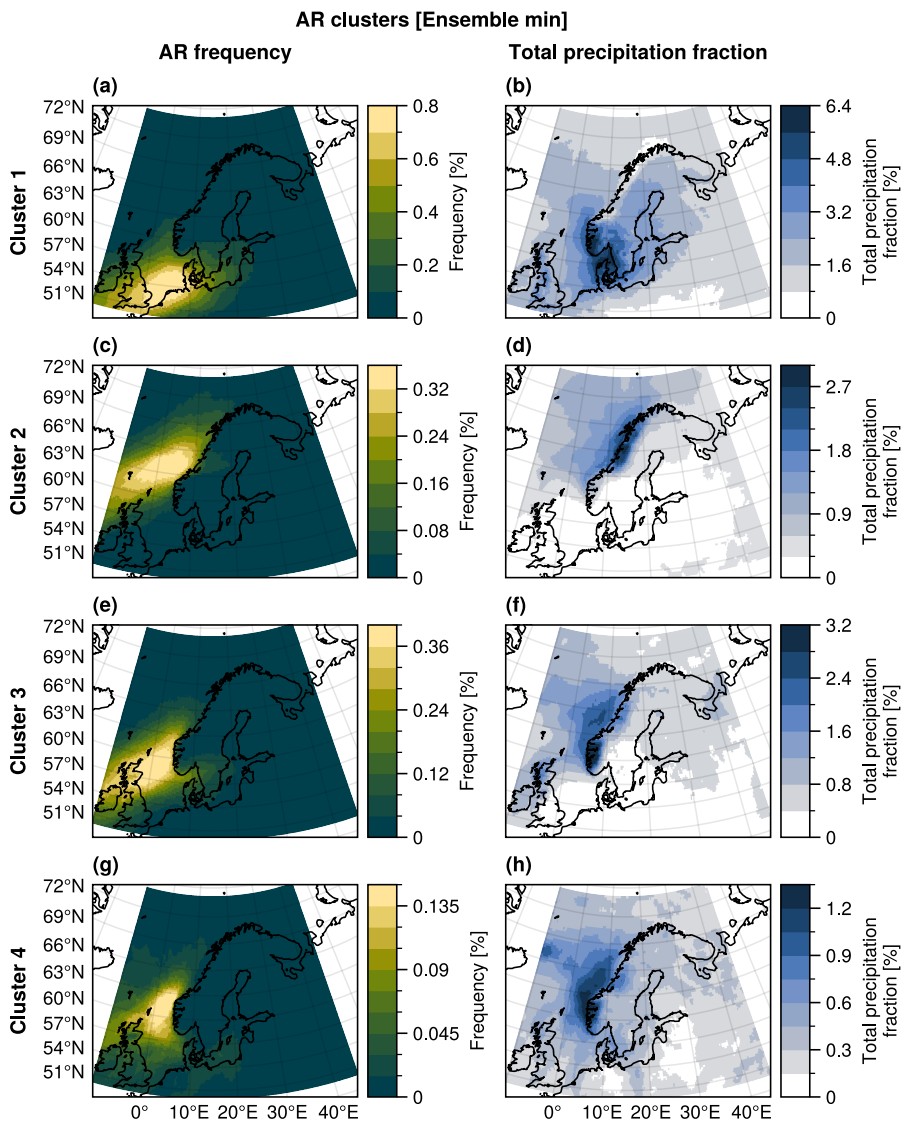

**Figure A3.** Ensemble minimum annual average AR frequencies (left column) and total precipitation fraction (right column) for each identified AR cluster (rows). Note that the colour scales differ between the subplots.



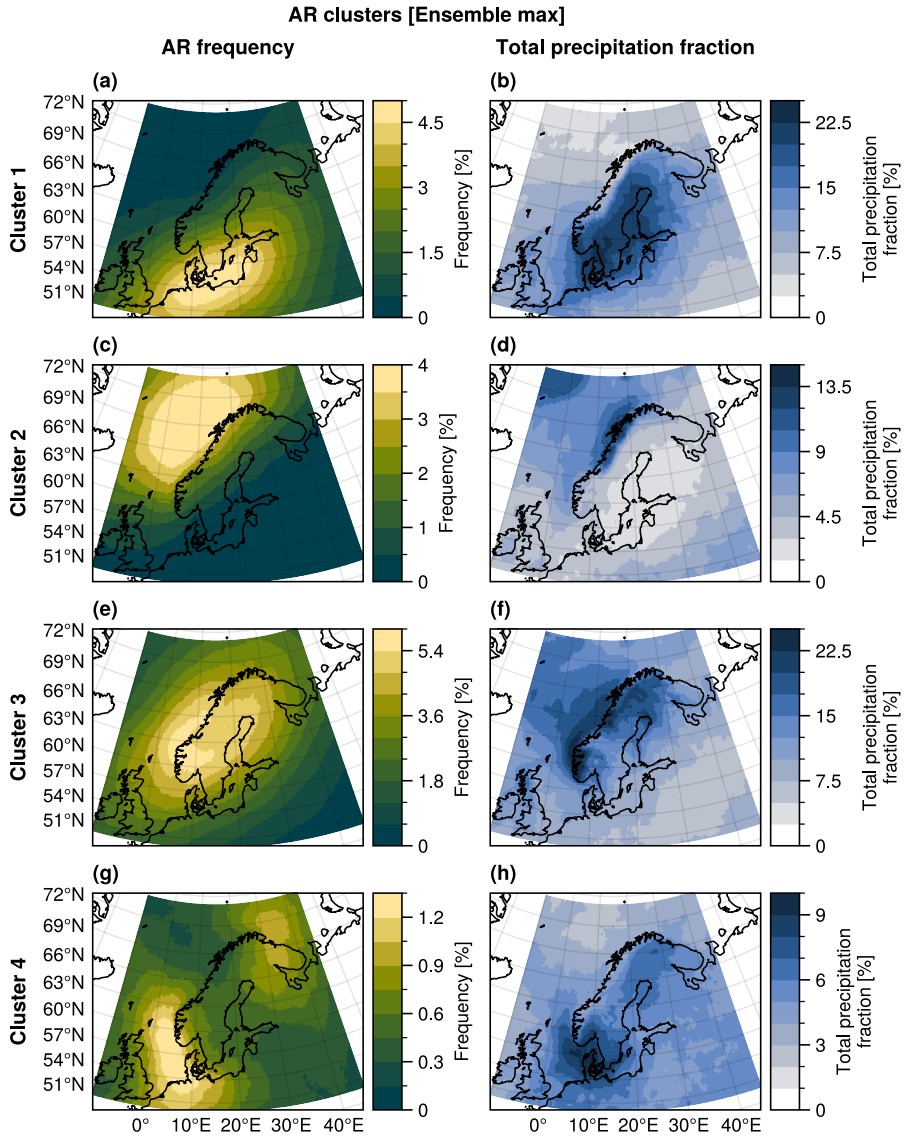

**Figure A4.** Ensemble maximum annual average AR frequencies (left column) and total precipitation fraction (right column) for each identified AR cluster (rows). Note that the colour scales differ between the subplots.

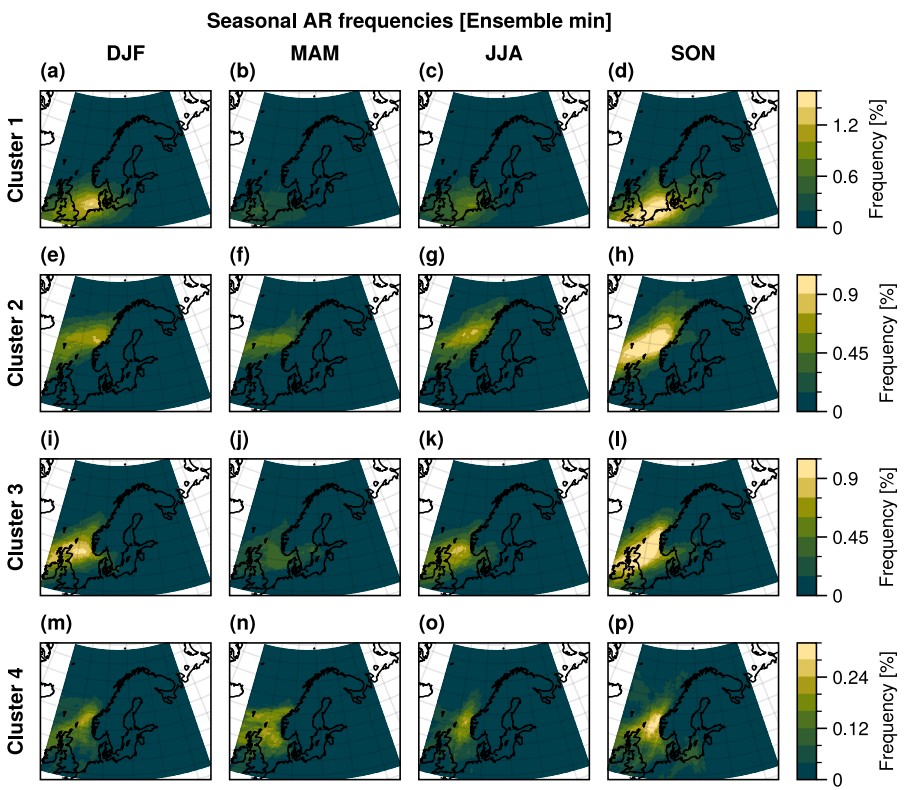

**Figure A5.** Maps of the average annual AR frequencies (ensemble minimum) during the seasons: winter (DJF), spring (MAM), summer (JJA), and autumn (SON) for each AR cluster. The frequencies correspond to the fraction between the number of AR time steps in each season and the total number of time steps in each respective season.

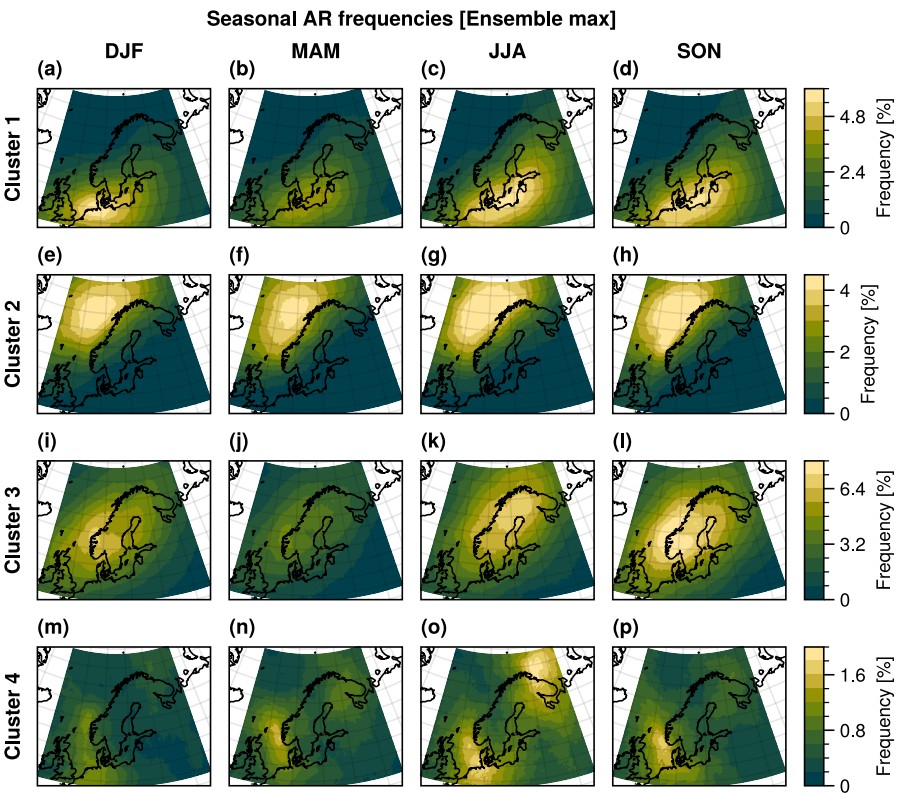

**Figure A6.** Maps of the average annual AR frequencies (ensemble maximum) during the seasons: winter (DJF), spring (MAM), summer (JJA), and autumn (SON) for each AR cluster. The frequencies correspond to the fraction between the number of AR time steps in each season and the total number of time steps in each respective season.

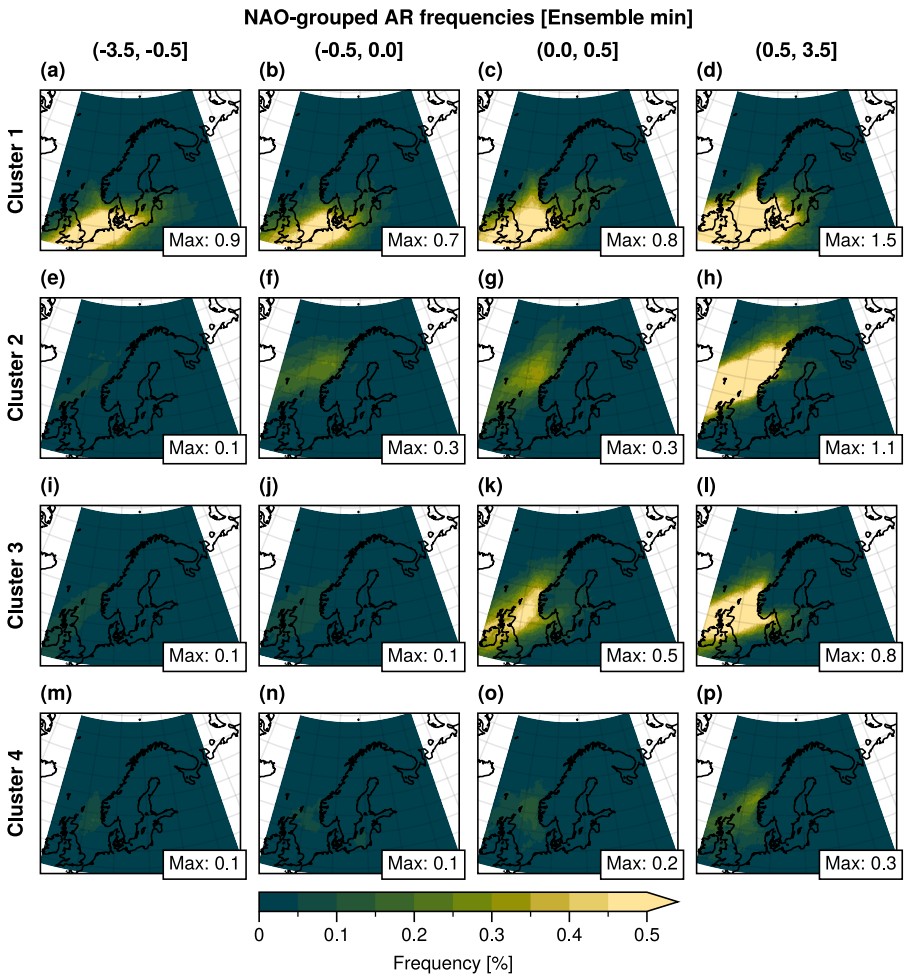

**Figure A7.** Maps of the average AR frequency (ensemble minimum) during the different phases of the North Atlantic Oscillation (NAO) for each AR cluster. The NAO phases are categorized into four groups: strongly negative (first column), weakly negative (second column), weakly positive (third column), and strongly positive (fourth column). The corresponding NAO index intervals are shown above each column. The colour scale is capped at 0.5 %; the maximum frequency value for each subplot is displayed in the bottom right corner.

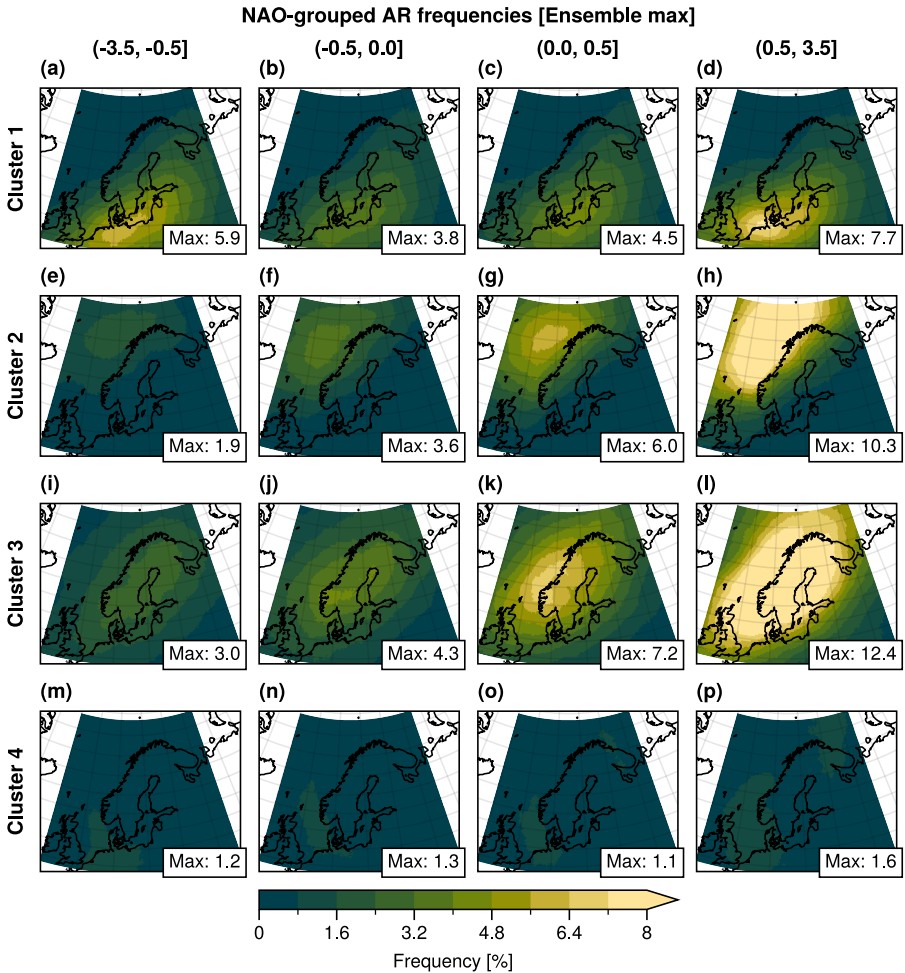

**Figure A8.** Maps of the average AR frequency (ensemble maximum) during the different phases of the North Atlantic Oscillation (NAO) for each AR cluster. The NAO phases are categorized into four groups: strongly negative (first column), weakly negative (second column), weakly positive (third column), and strongly positive (fourth column). The corresponding NAO index intervals are shown above each column. The colour scale is capped at 8 %; the maximum frequency value for each subplot is displayed in the bottom right corner.

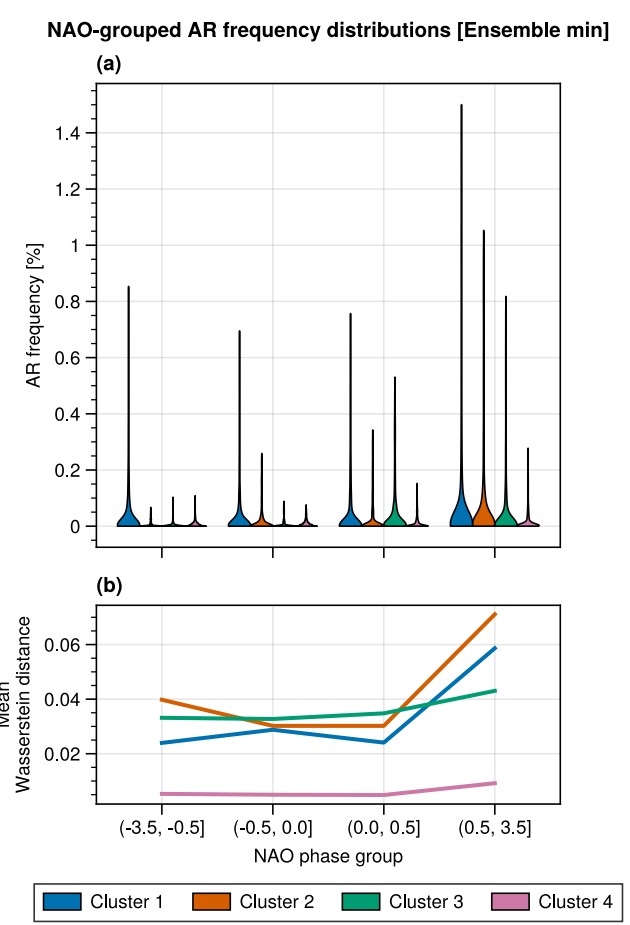

**Figure A9.** AR frequency distributions grouped by **(a)** NAO phase and **(b)** the mean Wasserstein distance between the successive distributions for each AR cluster (ensemble minimum). The distributions in **(a)** are visualized using violin plots. The height of the violin covers the whole range of values in the distribution, while the width at any given value indicates the relative frequency of that value in the distribution.

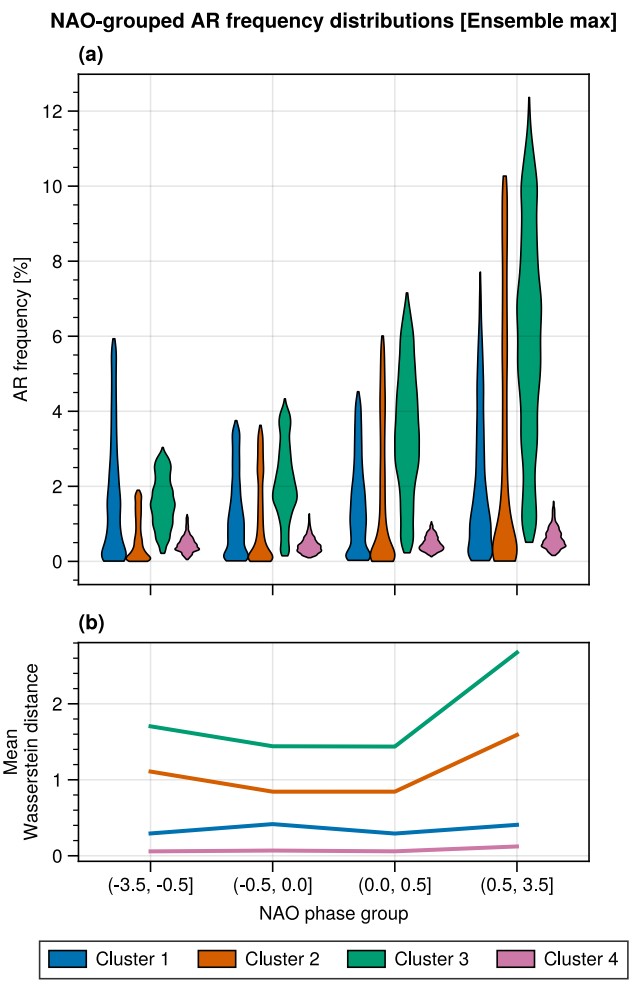

**Figure A10.** AR frequency distributions grouped by **(a)** NAO phase and **(b)** the mean Wasserstein distance between the successive distributions for each AR cluster (ensemble maximum). The distributions in **(a)** are visualized using violin plots. The height of the violin covers the whole range of values in the distribution, while the width at any given value indicates the relative frequency of that value in the distribution.



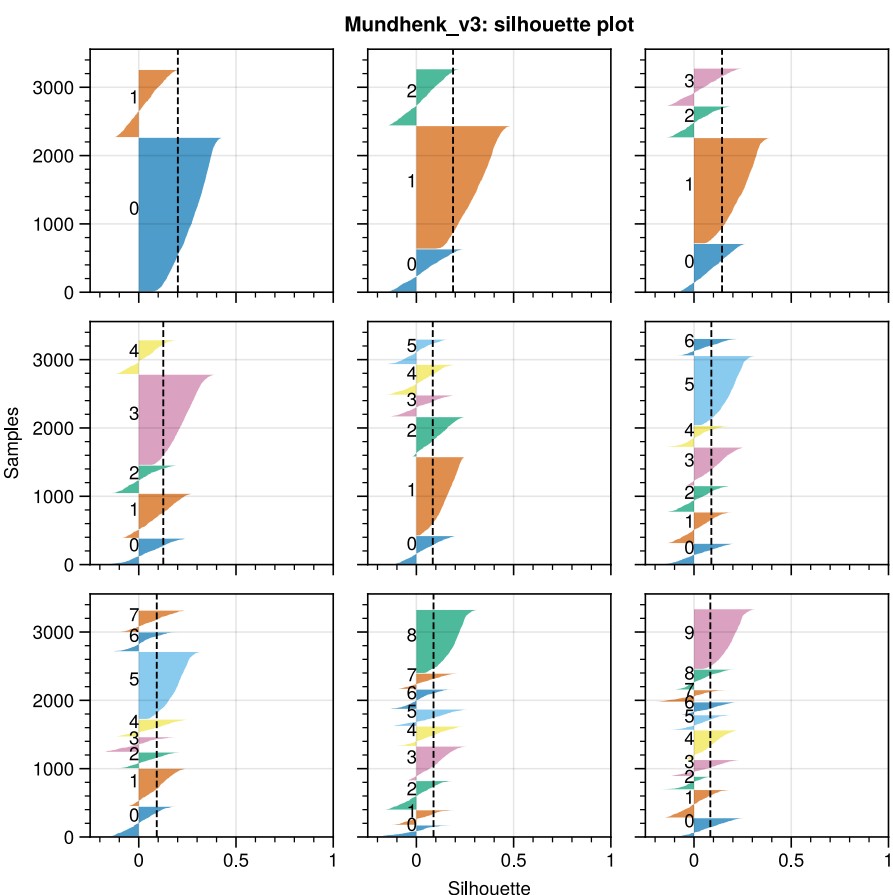

**Figure A11.** Silhouette plots for the Mundhenk_v3 ARDT. Each panel holds the silhouette for the respective number of clusters.



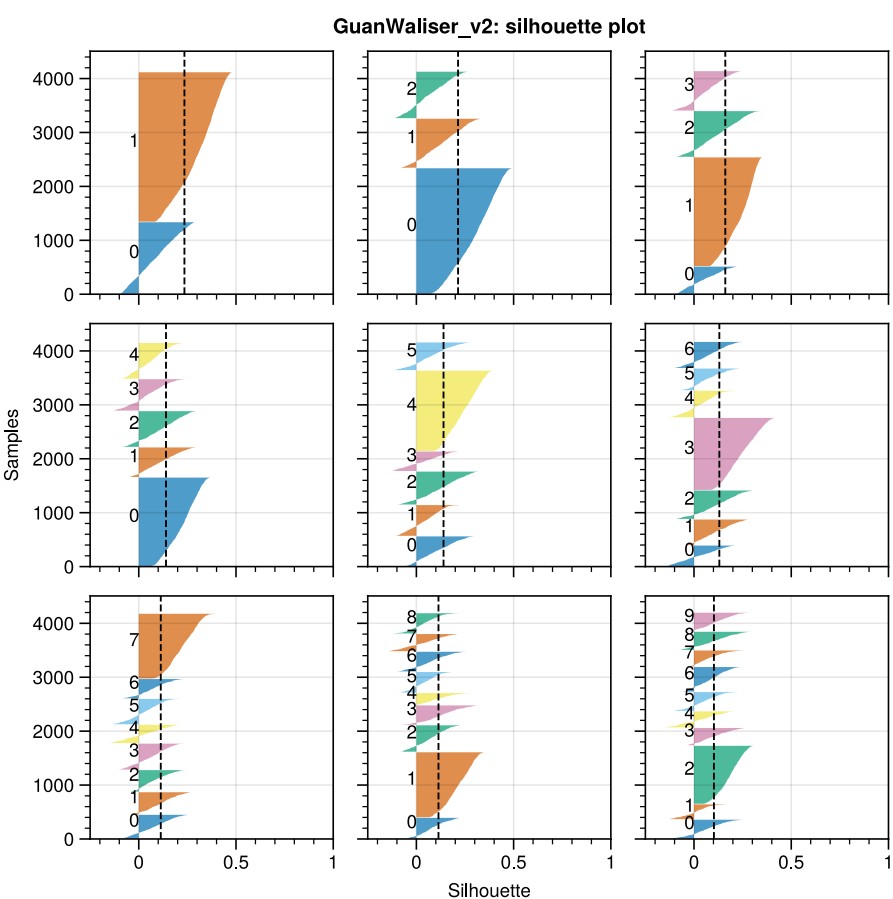

**Figure A12.** Silhouette plots for the GuanWaliser_v2 ARDT. Each panel holds the silhouette for the respective number of clusters.

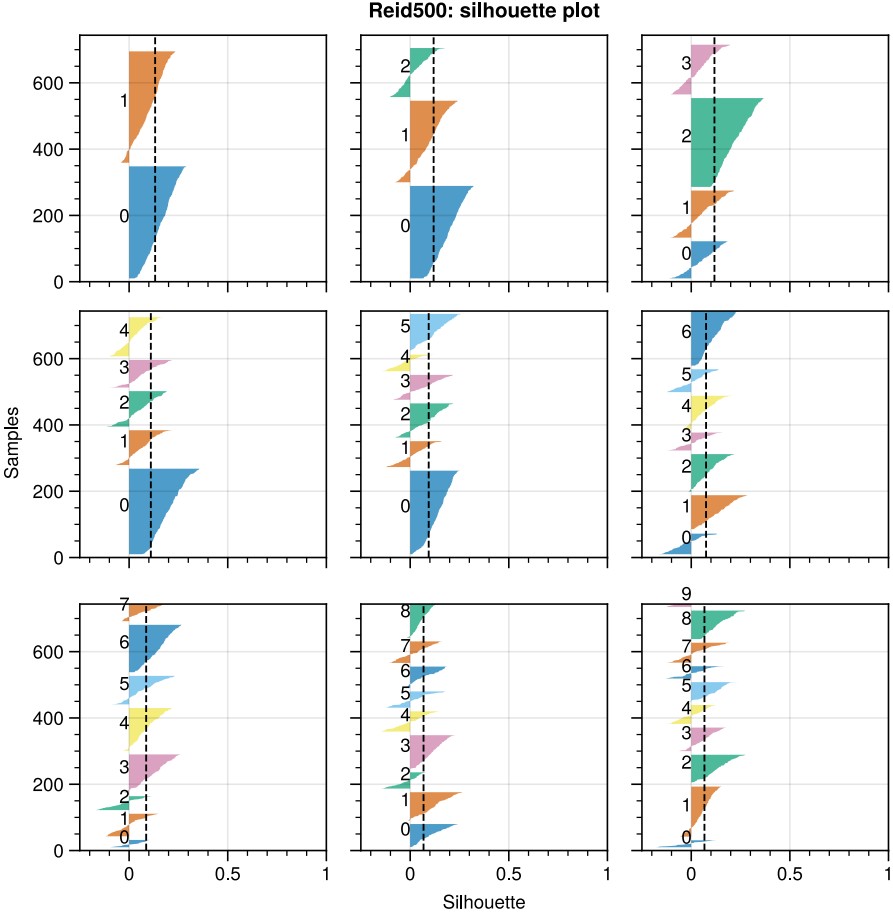

**Figure A13.** Silhouette plots for the Reid500 ARDT. Each panel holds the silhouette for the respective number of clusters.



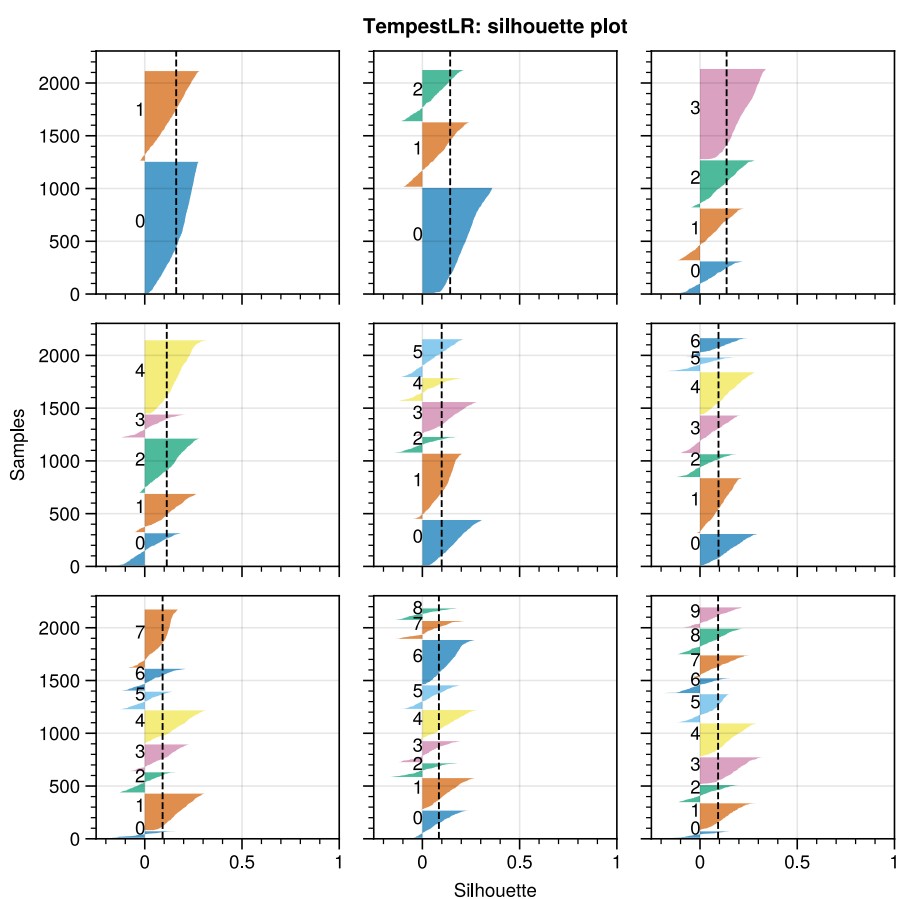

**Figure A14.** Silhouette plots for the TempestLR ARDT. Each panel holds the silhouette for the respective number of clusters.



*Author contributions.*  EH initiated the study, developed and performed the analysis, and wrote the manuscript. HWC supervised the project, contributed to the study design and interpretation of the results, and critically revised the manuscript.

*Competing interests.*  The authors have no competing interests to declare.

*Acknowledgements.*  This study was funded by internal funding from Chalmers University of Technology. We thank the ARTMIP project for providing open access to AR detection data, and the European Centre for Medium-Range Weather Forecasts (ECMWF) for making the ERA5 reanalysis data openly available. Finally, we would like to acknowledge the Python scientific community, and notably the open source libraries Dask and Xarray, which have been of great use in this study.



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
