# Peer review of "A climatology of atmospheric rivers over Scandinavia and associated precipitation"

_EGUsphere, 2025_

## Editor Comment (EC1)

**Referee Comments of "A climatology of atmospheric rivers over Scandinavia and related precipitation" by *Erik Holmgren and Hans W. Chen**

The manuscript studies the climatology of atmospheric rivers (ARs) in Scandinavia. The authors investigate the frequency, seasonality, and associated precipitation of four different pathways through which ARs reach Scandinavia, identified using k-means clustering. Additionally, they examine the relationship between large-scale atmospheric configurations (through the NAO index) and AR occurrences in the region. The manuscript is well written and well structured, and it addresses a relevant topic that has not been previously studied for the Scandinavian region. The manuscript is suitable for publication in *WCD* after some methodological clarifications.

- 1. The choice of ARDTs (Section 2.1): The ARTMIP project offers a large number of ARDTs. Why were these four chosen? Regarding the choice of ARDTs, there are other considerations to take into account:
  - a. TempestLR algorithm (*Table 1*): The threshold used is based on the Laplacian of IVT. The threshold of 250 kg·m-1·s-1 stated there is probably not correct (Ullrich et al., 2021). Please make sure to include the correct threshold used in the catalogue.
  - b. Reid500 algorithm: Using a fixed threshold of 500 kg·m-1·s-1 does not seem very suitable for Scandinavia, where some regions lie within the polar circle and these values of IVT might be too high. There are ARDTs specifically tailored for ARs in polar regions; perhaps one of those would be more appropriate for this study (Gorodetskaya et al., 2014).
  - c. Line 324: Uncertainties regarding ARDTs are presented here. It would be nice if the authors could provide a recommendation of which of ARDTs are more suitable for this region, which lies between the mid-latitudes and polar regions, as no ARDTs are specifically designed for it.
- 2. In Lines 105–107, it is described how ARs are tracked, with the tracking condition being "4-connected component labeling". It is not clear what this means. Please expand this explanation to make it clearer. Could it mean that an AR should be detected through four consecutive time steps?
- 3. The AR tracking condition mentioned in the previous comment is applied to the detected blobs from the ARDTs in the catalogues, which themselves do not track ARs, this is not obvious since some of these ARDTs also track ARs. You mention this in Lines 334–336; this should be stated at this point as well.
- 4. In Line 113, the domain where the AR tracking is applied is introduced. It could be presented earlier (even in the previous Data Section), as this would improve the understanding of the methodology for readers who are not familiar with AR detection and tracking procedures. However, applying the tracking threshold to such a small domain might lead to missing ARs that reach the area for fewer than four time-steps. Considering a larger domain when applying the AR tracking condition could help prevent this issue. In *Figure 2a*, the Scandinavian AR frequency is shown, with a maximum around the west coast of Denmark, and the frequency decreases considerably around the British Isles, which typically exhibit a higher AR frequency (Thandlam et al., 2022; Guan et al., 2015) than the North Sea. Please ensure that this pattern

- is indeed due to the condition that only ARs intersecting Scandinavia are included, and not an artefact of applying the AR-tracking condition within too small a domain.
- 5. In *Line 153*, it is explained how the AR precipitation is calculated, which is done by subtracting the non-AR precipitation. However, how is the non-AR precipitation calculated? I assume this is done when computing the field correlations between AR frequency and precipitation patterns. This process is not very clear, please clarify and improve the explanation in this section.
- 6. The NAO bins (*Lines 168–171*) are defined in four groups of equal size centered at 0. Nonetheless, this choice seems somewhat arbitrary and, later in the Results section, appears to limit the interpretation of the results (as all clusters seem to be mostly associated with a "strong positive" NAO). Since the NAO index is used as an indicator of large-scale patterns, the bins defined here might not be representative of specific large-scale configurations. For example, from 0 to 0.5 it is defined as a positive NAO, and between 0.5 and 3.5 as a strong positive NAO, but a value of 0.2 might not be representative of a positive NAO pattern, and a value of 0.7 might not represent a strong positive NAO, whereas a value of 2 likely would. Have you investigated whether your results are sensitive to the choice of these bins? One could consider defining NAO bins that are more physically relevant for certain weather patterns (e.g., Song et al., 2014, use a different threshold for NAO bins) or applying a weather regime approach (e.g., as in Messori et al., 2023), which goes beyond the NAO index.

**References:**

Ullrich, P. A., Zarzycki, C. M., McClenny, E. E., Pinheiro, M. C., Stansfield, A. M., and Reed, K. A.: TempestExtremes v2.1: a community framework for feature detection, tracking, and analysis in large datasets, Geosci. Model Dev., 14, 5023–5048, https://doi.org/10.5194/gmd-14-5023-2021, 2021.

Gorodetskaya, I. V., M. Tsukernik, K. Claes, M. F. Ralph, W. D. Neff, and N. P. M. Van Lipzig (2014), The role of atmospheric rivers in anomalous snow accumulation in East Antarctica, Geophys. Res. Lett., 41, 6199–6206, doi:10.1002/2014GL060881.

Thandlam, V., Rutgersson, A. & Sahlee, E. Spatio-temporal variability of atmospheric rivers and associated atmospheric parameters in the Euro-Atlantic region. Theor Appl Climatol 147, 13–33 (2022). <a href="https://doi.org/10.1007/s00704-021-03776-w">https://doi.org/10.1007/s00704-021-03776-w</a>

Guan, B., and D. E. Waliser (2015), Detection of atmospheric rivers: Evaluation and application of an algorithm for global studies, J. Geophys. Res. Atmos., 120, 12514–12535, doi:10.1002/2015JD024257.

Song, J., Li, C. & Zhou, W. High and low latitude types of the downstream influences of the North Atlantic Oscillation. Clim Dyn 42, 1097–1111 (2014). <a href="https://doi.org/10.1007/s00382-013-1844-3">https://doi.org/10.1007/s00382-013-1844-3</a>

Messori, G., & Dorrington, J. (2023). A joint perspective on North American and Euro-Atlantic weather regimes. Geophysical Research Letters, 50, e2023GL104696. <a href="https://doi.org/10.1029/2023GL104696">https://doi.org/10.1029/2023GL104696</a>

---

## Author Comment (AC1)

**EGUSPHERE-2025-3992: Authors Response**

**Erik Holmgren and Hans W. Chen**

**1 Response to referee 1**

 $https: \=//doi.org/10.5194/egusphere-2025-3992\text{-}RC1$

We thank referee 1 for taking the time to assess our manuscript and providing valuable feedback. In this document, we have included said feedback along with our responses in blue.

The study "A climatology of atmospheric rivers over Scandinavia and related precipitation" by Holmgren and Chen provides an analysis of atmospheric rivers (ARs) in Scandinavia. Here, the authors use four detection algorithms (three based on relative thresholds and one using an absolute threshold), with most of the results showing the ensemble median of these algorithms. Additionally, they investigate AR-related precipitation using ERA5 reanalysis, define different AR pathways by using a k-means clustering approach, and analyze the relationship between AR activity and the North Atlantic Oscillation (NAO) index.

Overall, it is an informative study that gives an overview of ARs and their associated precipitation over Scandinavia, particularly highlighting their strong influence on the west coast of Norway. The manuscript is generally well written; however, certain parts – especially Methodology 2.2 – would benefit from better explanations. For example, including a schematic related to the Jaccard index would greatly enhance clarity.

I would recommend this manuscript for publication after addressing the following points:

• For the detection of ARs, you have used the Reid500 detection algorithm. For me, the use of this algorithm is a bit surprising, as the absolute threshold of 500 kg m-1 s-1 seems to be high for this region. Could you explain why you chose this algorithm, or why you did not consider lowering the threshold?

That is a good question, and maybe we are not clear enough on this in the manuscript, but we have not run any of the ARDTs ourselves. The analysis is all based on data that is published through the ARTMIP project. We have simply used all available results from tracking algorithms that have a global coverage and cover the entire period from 1980 to 2019 for the ERA5 dataset. The limitations of the Reid500 algorithm for high latitudes are discussed briefly on L319. We will clarify that we are not running the ARDTs in section 2.1 and expand the discussion on the limitations

of the Reid500 algorithm.

• Section 2.2: You have used the Jaccard index to analyse the temporal relationship between AR objects. In Line 121, you refer to the "similarity" of AR objects. Could you clarify what kind of similarity you are referring to? Is it limited to the IVT amount, or are other variables, such as moisture, wind, or temperature, also considered? Furthermore, in Lines 127-130, it seems that some AR objects may be excluded (I'm not sure if I understand this correctly). If this is the case, would this not result in an underestimation of AR occurrence?

The Jaccard index measures the morphological similarity between two AR objects, so it does not take any of the variables you mention into account. It is instead purely a measure of how similar in shape the two AR objects are. This is the same measure used for the tracking of ARs in the algorithm from Guan and Waliser. On L127-L130 we describe how individual AR objects are tracked through time and how the best match is selected. If no matching AR object is found, the AR object is still kept as an AR that spans a single time step, so no ARs are excluded. We will add a sentence that clarifies this.

• Section 2.3: Here, you use the k-means clustering to define different AR pathways. The terminology in this section is a bit confusing. First, you use the term "clustering", but then you switch to "pathways". I recommend using "pathway" consistently throughout the text. It would also be helpful to include a table summarizing the cluster/pathways.

It is very good that you bring this up. When we describe the method, we talk about clustering as this is a widely adopted term for identifying clusters in data of any sort. The results from any clustering are the clusters – essentially groups of data points. In our specific case we have then chosen to refer to the resulting clusters as AR pathways, as we mention on L149-150, as this is a better physical description of the groups (clusters) of ARs. But we agree that we are currently not consistent in the use of the terms AR pathways and clusters. We will update the paper, including the figures, to consistently use the term "AR pathways", except when we introduce the clustering method.

**Minor comments:**

L6-7: Clarify which regions within Scandinavia are most active

Good suggestion, we will do this.

L7: Specify that the 40% precipitation refers to local values

We will make this change.

L14: Which clusters? Do you mean pathways?

Yes, good catch. This will be changed along with other changes regarding the use of AR pathways.

L23-24: Add a reference supporting that ARs are important for Scandinavia

This sentence refers to the potential importance of ARs in Scandinavia, as it is not really known if they are important yet. But to clarify this, we will expand the part about that the region is relatively understudied when it comes to ARs.

L31-47: The overview of ARs in other regions is useful but could be condensed, since these regions are not compared further in the manuscript.

We think that this section is valuable to set the context of ARs in the climate system, and why it should be interesting to investigate their impacts over Scandinavia. However, we will work to condense it somewhat.

L79-84: I would structure the manuscript in 3 main sections: 2 Methods, 3 Results, 4 Summary.

We will add a main "Results and Discussion" section header, with our previous result sections as subsections. This will make it more clear when the results start and improve the overall organization.

L93: dived  $\rightarrow$  divided

We will correct this. Thanks!

L93: absolute IVT threshold or a relative IVT threshold  $\rightarrow$  absolute or relative IVT threshold

We will adopt this.

L181: Clarify that the precipitation values refer to local contributions

The result is specified as a fraction of the annual average local precipitation. But we will restructure this sentence to make this more clear.

L181: Here, you refer to Scandinavia as a region, but in the next sentence, you write: "specifically, the regions with the highest AR-related precipitation". Here, I would suggest specifying subregions more clearly. For example, the southeast coast of Norway and the west coast of Denmark are mostly affected by AR-related precipitation.

Good suggestion, we will incorporate this. Thanks!

Figure 4: Do you have any idea why the spread is higher for smaller precipitation rates?

This is a good question. We think that this could have to do with the intensity of the ARs and how/if they are identified by all ARDTs: The less intense ARs are likely not detected as often by the ARDTs with high IVT thresholds. At the same time, these less intense ARs are typically associated with weaker precipitation.

Figure 5: I see why you have used different color scales, but you have to be very careful when comparing them with each other. Maybe you could use a gradient colormap with more colors

Yes, this is intentional, and we take this into account when comparing the patterns by referencing the absolute values of the plots. Using the same colour scale for all plots would make them harder to read, as the features in e.g. Figure 5g would not be visible if using the scale of Figure 5a. To make it even more clear we will add the max values as an annotation (similar to Fig. 8) to the subplots in this and other figures showing maps.

Figure 5: It is interesting to see the high precipitation fraction in Northern Sweden. I would suggest seeing it more in cluster 3

Yes! The precipitation fraction in cluster 3 is in absolute terms higher than in cluster 2, but the precipitation pattern is more defined in Northern Sweden and Norway.

Line246-257: The described seasonal pattern does not apply to Cluster 4, where ARs are less active in winter and more active in summer

That is correct. We will update the text to reflect this.

L265: Consider rephrasing: Following autumn, ARs are less frequent during winter and reach their maximum in spring.

Good suggestion, we will incorporate this.

L270: For Cluster/pathway 4: SON looks quite similar to MAM. It seems that there is already a decrease in autumn compared to summer

Yes, you are right. We will remove "and autumn (SON)" from this sentence. Thanks!

**2 Response to referee 2**

https://doi.org/10.5194/egusphere-2025-3992-RC2

We thank referee 2 for taking the time to assess our manuscript and providing valuable feedback. In this document, we have included said feedback along with our responses in blue.

**SUMMARY**

This manuscript presents an analysis of the typical frequency and precipitation impacts of atmospheric rivers (ARs) in Scandinavia. The authors find that from 1980-2019, ARs contributed 40% of the total annual precipitation while only occurring 5% of the time – elevating their role as a primary driver for extreme precipitation in the region. There is large spatial variability in the precipitation impacts of ARs in the region, which is closely related to four key pathways of ARs and their proximity to topography upon landfall. They also identified peaks in AR activity during strong positive phases of the North Atlantic Oscillation.

The study is novel, robust, interesting, and organized in a logical manner. It is well-written and advances our understanding of how the latitudinal positioning and orientation of ARs affects their precipitation impacts. I have included a few major comments and some minor comments, organized by line number, below.

**MAJOR COMMENTS**

I noticed that the data analysis spans 1980-2019. I am curious if there is a specific reason why the study is not extended to a more recent year, such as 2024. Is this a limitation of the available AR detection tools? I see it as advantageous to include as many years as possible in the data analysis, particularly recent years, as these have been marked by events that are more extreme than we typically observe in the 1980s and 1990s.

We fully agree that it would be beneficial to have the study extended up to 2024. This limitation exists because the ARTMIP data we use only extends up to 2019. We will clarify that we have used data from the ARTMIP project, and its inherent limitations, in section 2.1.

Is there a reason why you do not separate the total precipitation into rainfall and snow-fall explicitly, as these can have different impacts on the surface and long-term implications for the hydrology of the region? (for example, L90).

This is a good question. For this study we are primarily investigating ARs and the total moisture they bring to Scandinavia. Analysing rainfall and snowfall separately would bring additional detail, but would extend the study significantly. So, we think this would be better suited for a separate study. We will add a comment about this when discussing possible future work in the conclusions.

For the NAO-AR pathway results, it may help readers to include a map of the typical atmospheric circulation during the positive and negative phase of the NAO (or include it as a supplementary figure).

This is a very good suggestion. We will add a map to describe the state of atmospheric circulation during the different phases of the NAO. Thanks!

**MINOR COMMENTS**

Title – "associated" might be more precise than "related"

Good suggestion. We will change this.

6 – "present during up to 5% of the 6-hourly timesteps in the most active areas" is a bit confusing. Does this mean they are present 5% of the time? Combined with "the region receives" in the following sentence, it is unclear which parts of Scandinavia are referred to, and whether the following sentence is talking about the hotspots for AR activity or all of Scandinavia.

Yes, this means that they are present up to 5% of the time. We mention it since we want to make it clear that we have been working with 6-hourly data. We will rephrase the sentence to clarify this. The part about that "the region receives" has also been mentioned by another reviewer, and we will rephrase this to be more explicit as to where this precipitation occurs.

8 - "probability density is greater for the highest precipitation rates during AR events" > can you rephrase and say "ARs are disproportionately associated with the highest precipitation rates"?

Thank you for the suggestion. We feel that "disproportionately" might overstate the differences that we observe, but we will revise the sentence to make it clearer.

Rest of the introduction (25-73) – I would recommend reworking this section to hone into the Scandinavian region and what is already known about ARs and precipitation more quickly. Currently, the text covers ARs and their precipitation impacts in a lot of different regions across the world, and I think this is probably more information than is needed to contextualize the study. Furthermore, I think it would motivate the results better to dig into the climatology of the region of study and familiarize readers with what is and isn't known about precipitation and specifically AR-driven precipitation here. It is currently addressed on lines 69-73, but this is brief compared to the rest of the introduction.

We mention the other regions in order to contextualize why looking into ARs over Scandinavia is interesting, as there currently is not that much known about their impacts over Scandinavia. However, we can see that this section is a bit long, and we will work to condense it.

79 – consider explicitly stating the names of the ARDTs used

Good suggestion, we will add them here.

88 – recommend mentioning the 0.25-degree grid spacing of ERA5 here

Good suggestion, we will include this.

89 – "to make the computation feasible" > "to improve computational efficiency"

We will adopt this.

96 – recommend explain the method for each AR detection tool in more detail. For example, the exact thresholds are listed in the table, but how might that lead to differences among the AR detection tools? Which one is more or less restrictive?

For a full description of the respective tool we think that it is better to refer to the cited papers. As for the differences in the results from the different ARDTs, we discuss this in section 7.

106 – do all of the AR catalogs have a requirement for continuity of the AR? As in, in some tools there can be a timestep where there are little bits of AR that show up as noise or are likely part of a larger AR but are disjoint. How does the 4-connected component labelling method handle these, if they are present?

The four ARDTs we have used all use some type of geometry filtering after the initial condition based identification. The GuanWaliser, Mundhenk and Reid ARDTs all require the ARs to be longer than a certain length (typically 1500-2000 km) and have a length to width ratio of at least 2. The Tempest ARDT is slightly different, instead requiring the connected area of an AR to be  $>\approx$ 125 000 km². Thus, there is minimal disjointed noise in the AR catalogs that we used. If there were significant noise, our algorithm would identify more AR objects. However, this would not affect the AR frequencies and would likely have little impact on the AR pathways.

111 – regarding the buffering, I think this approach makes a lot of sense! Wondering if there is a precedent for the method that you can reference, or if this is the first time this has been applied in this part of the world?

Thanks! We developed this approach independently and, as far as we are aware, there is no prior precedent.

Fig1 – I find this map very helpful. I would recommend adding the names of the different regions mentioned in the study to the map.

Good suggestion, we will add the names to the map.

144 – why do you think the number of optimal clusters was different for the four AR detection tools? What characteristic exerts the greatest influence in the clustering, and what does that suggest about the differences among the catalogs?

This is a good question, and also a tricky one. The biggest difference among the different ARDTs is the spatial extent of the ARs that they detect. With the larger ARs, the overlap

in the data will inherently be larger, which in turn will aid the clustering. So for the ARDTs detecting smaller ARs, we suspect that more clusters are needed to capture the variability among the ARs. But the resulting clusters are large enough to overlap so that we can combine them.

153 – do you believe that non-AR associated precipitation occurs during an AR event, in the footprint of the AR? Or is this because you're summing all precipitation in Scandinavia during AR landfall anywhere in the region?

Fully attributing the precipitation during an AR to different sources is tricky. We except that there can be non-AR associated precipitation in the AR footprint. This is why we subtract the climatological average of non-AR precipitation, as this should isolate the AR-specific contribution from the background precipitation.

177-179 – consider moving this to the methods

We see value in having it here to remind the reader of what they are going to look at.

180-185 – what is the standard deviation in the frequency and precipitation?

Since our ensemble only consists of four members, we think that the standard deviation will not be representative/meaningful. Instead, we present the minimum and maximum of the ensemble. We see that this is missing here and we will add it – thanks!

191 – add "frequency" > "annual AR frequency and precipitation"

Thanks! We will correct this.

206 – how did you decide on this cutoff for high precipitation amounts?

We infer this from panel b in Figure 4. It shows that the probability densities are higher during AR events for precipitation rates  $12 \text{ mm } 6h^{-1}$  and above. We will clarify this in the text.

209 – consider converting to hourly or daily rates, which may be more comparable across different studies, as opposed to mm per 6h

This is a valid point; we do however think that converting from the 6-hourly accumulated values to either mm/h or mm/day could be misleading. The distribution of hourly values is likely different from the 6-hourly values, while for the conversion to daily values we would have to assume that the 6-hourly precipitation persist throughout the day. Alternatively, if we further aggregate the 6-hourly values to daily, it would be nontrivial to assign the daily precipitation to AR events that last for less than a full day (e.g. half a day, or 2.25 days).

Therefore, we think the clearest approach is to work with the 6-hourly precipitation.

Fig5 – would it make sense to put 5a,c,e, and g on a common colorbar (value range for the colors)? And same for b,d,f, and h? Currently the total precipitation fraction for

5f looks much larger than any of the other panels, purely because of the dominance of dark blue colors, even though in reality it has smaller values than the other panels.

We have been going back and forth on this throughout the development of the study. The benefit of having the same colour scale for the subplots is, as you mention, that it would be more straightforward to compare the clusters to each other in terms of absolute magnitudes. However, due to the relatively large range, a lot of (if not all) the details in the less active clusters would be lost in the darker colours. So we have opted to have separate colour scales, but we are trying to make this clear by always referencing the maximum values in the text. To further clarify, we will add annotations showing the max value for each subplot, as is done in Fig. 8.

Fig6 – similar to the previous comment – you could normalize these so that the colorbar is the same and the percentage of ARs by season sums to 1.

See the response to the comment above. Additionally, we did think about some type of normalization for these plots, but we also wanted the numbers to be easy to interpret. For instance a frequency of 4% here corresponds directly to the fraction of time steps within that season. Any normalization would make the figure less intuitive and require an additional explanation of what is shown.

264-267 - could add these place names to Fig1

Yes, we will do this.

319 – "are typically lower" add "due to lower air temperatures"

Thanks! We will adopt this.

---

## Author Response (AR1)

**EGUSPHERE-2025-3992: Final Response**

Erik Holmgren and Hans W. Chen

**1 Response to referee 1**
**https://doi.org/10.5194/egusphere-2025-3992-RC1**

We thank referee 1 for taking the time to assess our manuscript and providing valuable feedback. In this document, we have included said feedback along with our responses in blue.

The study "A climatology of atmospheric rivers over Scandinavia and related precipitation" by Holmgren and Chen provides an analysis of atmospheric rivers (ARs) in Scandinavia. Here, the authors use four detection algorithms (three based on relative thresholds and one using an absolute threshold), with most of the results showing the ensemble median of these algorithms. Additionally, they investigate AR-related precipitation using ERA5 reanalysis, define different AR pathways by using a k-means clustering approach, and analyze the relationship between AR activity and the North Atlantic Oscillation (NAO) index.

Overall, it is an informative study that gives an overview of ARs and their associated precipitation over Scandinavia, particularly highlighting their strong influence on the west coast of Norway. The manuscript is generally well written; however, certain parts – especially Methodology 2.2 – would benefit from better explanations. For example, including a schematic related to the Jaccard index would greatly enhance clarity.

I would recommend this manuscript for publication after addressing the following points:

- For the detection of ARs, you have used the Reid500 detection algorithm. For me, the use of this algorithm is a bit surprising, as the absolute threshold of 500 kg m-1 s-1 seems to be high for this region. Could you explain why you chose this algorithm, or why you did not consider lowering the threshold?

  That is a good question, and maybe we are not clear enough on this in the manuscript, but we have not run any of the ARDTs ourselves. The analysis is all based on data that is published through the ARTMIP project. We have simply used all available results from tracking algorithms that have a global coverage and cover the entire period from 1980 to 2019 for the ERA5 dataset. The limitations of the Reid500 algorithm for high latitudes were discussed briefly on L319. In the revised manuscript, we have clarified that we are not running the ARDTs in section 2.1 (L80–84) and

expanded the discussion on the limitations of the Reid500 algorithm (L315–330).

- Section 2.2: You have used the Jaccard index to analyse the temporal relationship between AR objects. In Line 121, you refer to the "similarity" of AR objects. Could you clarify what kind of similarity you are referring to? Is it limited to the IVT amount, or are other variables, such as moisture, wind, or temperature, also considered? Furthermore, in Lines 127-130, it seems that some AR objects may be excluded (I'm not sure if I understand this correctly). If this is the case, would this not result in an underestimation of AR occurrence?

  The Jaccard index measures the morphological similarity between two AR objects, so it does not take any of the variables you mention into account. It is instead purely a measure of how similar in shape the two AR objects are. This is the same measure used for the tracking of ARs in the algorithm from Guan and Waliser. If no matching AR object is found, the AR object is still kept as an AR that spans a single time step, so no ARs are excluded. On L122–128 in the revised manuscript, we have updated the description of how individual AR objects are tracked through time and how the best match is selected.

- Section 2.3: Here, you use the k-means clustering to define different AR pathways. The terminology in this section is a bit confusing. First, you use the term "clustering", but then you switch to "pathways". I recommend using "pathway" consistently throughout the text. It would also be helpful to include a table summarizing the cluster/pathways.

  It is very good that you bring this up. When we describe the method, we talk about clustering as this is a widely adopted term for identifying clusters in data of any sort. The results from any clustering are the clusters – essentially groups of data points. In our specific case we have then chosen to refer to the resulting clusters as AR pathways, as we mention on L149–150 (Revised: L147), as this is a better physical description of the groups (clusters) of ARs. But we agree that we were not consistent in the use of the terms AR pathways and clusters. We have updated the manuscript, including the figures, to consistently use the term "AR pathways", except when we introduce the clustering method.

**Minor comments:**

L6-7: Clarify which regions within Scandinavia are most active

Good suggestion, we have fixed this on L6–7.

L7: Specify that the 40% precipitation refers to local values

We have clarified that this refers to the most active areas over Denmark and southern Norway on L7.

L14: Which clusters? Do you mean pathways?

Yes, good catch. We have changed this, and other instances of AR pathways in the revised manuscript.

L23-24: Add a reference supporting that ARs are important for Scandinavia

This sentence refers to the potential importance of ARs in Scandinavia, as it is not really known if they are important yet. We have updated the L23–24 to clarify this.

L31-47: The overview of ARs in other regions is useful but could be condensed, since these regions are not compared further in the manuscript.

We think that this section is valuable to set the context of ARs in the climate system, and why it should be interesting to investigate their impacts over Scandinavia. However, we have condensed it in the revised manuscript, notably L31–45.

L79-84: I would structure the manuscript in 3 main sections: 2 Methods, 3 Results, 4 Summary.

We have added a main "Results and discussion" section header, with our previous result sections as subsections.

L93: dived → divided

We have corrected this on L89. Thanks!

L93: absolute IVT threshold or a relative IVT threshold → absolute or relative IVT threshold

We have included this in the revised manuscript on L89.

L181: Clarify that the precipitation values refer to local contributions

The result is specified as a fraction of the annual average local precipitation. We have restructured this sentence to make this clearer (see L180 in revised manuscript).

L181: Here, you refer to Scandinavia as a region, but in the next sentence, you write: "specifically, the regions with the highest AR-related precipitation". Here, I would suggest specifying subregions more clearly. For example, the southeast coast of Norway and the west coast of Denmark are mostly affected by AR-related precipitation.

Good suggestion, we have incorporated this on L181. Thanks!

Figure 4: Do you have any idea why the spread is higher for smaller precipitation rates?

This is a good question. We think that this could have to do with the intensity of the ARs and how/if they are identified by all ARDTs: The less intense ARs are likely not detected as often by the ARDTs with high IVT thresholds. At the same time, these less intense ARs are typically associated with weaker precipitation.

Figure 5: I see why you have used different color scales, but you have to be very careful when comparing them with each other. Maybe you could use a gradient colormap with more colors

Yes, this is intentional, and we take this into account when comparing the patterns by referencing the absolute values of the plots. Using the same colour scale for all plots would make them harder to read, as the features in e.g. Figure 5g would not be visible if using the scale of Figure 5a. To make it even more clear we have added the max values as an annotation (similar to Fig. 8) to the subplots in this and other figures showing maps.

Figure 5: It is interesting to see the high precipitation fraction in Northern Sweden. I would suggest seeing it more in cluster 3

Yes! The precipitation fraction in cluster 3 is in absolute terms higher than in cluster 2, but the precipitation pattern is more defined in Northern Sweden and Norway.

Line246-257: The described seasonal pattern does not apply to Cluster 4, where ARs are less active in winter and more active in summer

That is correct. We have updated L245 to reflect this.

L265: Consider rephrasing: Following autumn, ARs are less frequent during winter and reach their maximum in spring.

Good suggestion, we have incorporated this on L264.

L270: For Cluster/pathway 4: SON looks quite similar to MAM. It seems that there is already a decrease in autumn compared to summer

Yes, you are right. We have removed "and autumn (SON)" from this sentence (L269). Thanks!

**2 Response to referee 2**

**https://doi.org/10.5194/egusphere-2025-3992-RC2**

We thank referee 2 for taking the time to assess our manuscript and providing valuable feedback. In this document, we have included said feedback along with our responses in blue.

**SUMMARY**

This manuscript presents an analysis of the typical frequency and precipitation impacts of atmospheric rivers (ARs) in Scandinavia. The authors find that from 1980-2019, ARs contributed 40% of the total annual precipitation while only occurring 5% of the time – elevating their role as a primary driver for extreme precipitation in the region. There is large spatial variability in the precipitation impacts of ARs in the region, which is closely related to four key pathways of ARs and their proximity to topography upon landfall. They also identified peaks in AR activity during strong positive phases of the North Atlantic Oscillation.

The study is novel, robust, interesting, and organized in a logical manner. It is well-written and advances our understanding of how the latitudinal positioning and orientation of ARs affects their precipitation impacts. I have included a few major comments and some minor comments, organized by line number, below.

**MAJOR COMMENTS**

I noticed that the data analysis spans 1980-2019. I am curious if there is a specific reason why the study is not extended to a more recent year, such as 2024. Is this a limitation of the available AR detection tools? I see it as advantageous to include as many years as possible in the data analysis, particularly recent years, as these have been marked by events that are more extreme than we typically observe in the 1980s and 1990s.

We fully agree that it would be beneficial to have the study extended up to 2024. This limitation exists because the ARTMIP data we used only extends up to 2019. We have clarified that we have used data from the ARTMIP project, and its inherent limitations, in section 2.1 (see L80–85).

Is there a reason why you do not separate the total precipitation into rainfall and snowfall explicitly, as these can have different impacts on the surface and long-term implications for the hydrology of the region? (for example, L90).

This is a good question. For this study we are primarily investigating ARs and the total moisture they bring to Scandinavia. Analysing rainfall and snowfall separately would bring additional detail, but would extend the study significantly. So, we think this would be better suited for a separate study. We have added a comment about this when discussing possible future work in the conclusions (See L385–386).

For the NAO-AR pathway results, it may help readers to include a map of the typical atmospheric circulation during the positive and negative phase of the NAO (or include it as a supplementary figure).

This is a very good suggestion. Thanks! We have reworked Fig. 8 to include maps of the daily average 500-hPa geopotential height (anomalies) during the different NAO phases in the revised manuscript.

**MINOR COMMENTS**

Title – "associated" might be more precise than "related"

Good suggestion. We have changed the title of the manuscript.

6 – "present during up to 5% of the 6-hourly timesteps in the most active areas" is a bit confusing. Does this mean they are present 5% of the time? Combined with "the region receives" in the following sentence, it is unclear which parts of Scandinavia are referred to, and whether the following sentence is talking about the hotspots for AR activity or all of Scandinavia.

Yes, this means that they are present up to 5% of the time. We mentioned it since we wanted to make it clear that we have been working with 6-hourly data. But we see that this might have been unnecessary and have removed the mention of 6-hourly time steps, and rephrased the sentence (L6–7). The part about that "the region receives" was also mentioned by another reviewer, and we have rephrased this to be more explicit as to where this precipitation occurs (L7).

8 - "probability density is greater for the highest precipitation rates during AR events" > can you rephrase and say "ARs are disproportionately associated with the highest precipitation rates"?

Thank you for the suggestion. We feel that "disproportionately" might overstate the differences that we observe, but we have clarified this sentence (L8–9).

Rest of the introduction (25-73) – I would recommend reworking this section to hone into the Scandinavian region and what is already known about ARs and precipitation more quickly. Currently, the text covers ARs and their precipitation impacts in a lot of different regions across the world, and I think this is probably more information than is needed to contextualize the study. Furthermore, I think it would motivate the results better to dig into the climatology of the region of study and familiarize readers with what is and isn't known about precipitation and specifically AR-driven precipitation here. It is currently addressed on lines 69-73, but this is brief compared to the rest of the introduction.

We mention the other regions in order to contextualize why looking into ARs over Scandinavia is interesting, as there currently is not that much known about their impacts

over Scandinavia. However, we can see that this section was a bit long, and we have condensed it, notably L31–45.

79 – consider explicitly stating the names of the ARDTs used

Good suggestion, we have added them to L70–71.

88 – recommend mentioning the 0.25-degree grid spacing of ERA5 here

Good suggestion, we added this on L82.

89 – "to make the computation feasible" > "to improve computational efficiency"

We have adopted this on L85.

96 – recommend explain the method for each AR detection tool in more detail. For example, the exact thresholds are listed in the table, but how might that lead to differences among the AR detection tools? Which one is more or less restrictive?

For a full description of the respective tool we think that it is better to refer to the cited papers. As for the differences in the results from the different ARDTs, we have expanded the discussion on this in section 3.5, notably L313–330.

106 – do all of the AR catalogs have a requirement for continuity of the AR? As in, in some tools there can be a timestep where there are little bits of AR that show up as noise or are likely part of a larger AR but are disjoint. How does the 4-connected component labelling method handle these, if they are present?

The four ARDTs we have used all use some type of geometry filtering after the initial condition based identification. The GuanWaliser, Mundhenk and Reid ARDTs all require the ARs to be longer than a certain length (typically 1500-2000 km) and have a length to width ratio of at least 2. The Tempest ARDT is slightly different, instead requiring the connected area of an AR to be $>\approx 125\,000$ km$^2$. Thus, there is minimal disjointed noise in the AR catalogs that we used. If there were significant noise, our algorithm would identify more AR objects. However, this would not affect the AR frequencies and would likely have little impact on the AR pathways.

111 – regarding the buffering, I think this approach makes a lot of sense! Wondering if there is a precedent for the method that you can reference, or if this is the first time this has been applied in this part of the world?

Thanks! We developed this approach independently and, as far as we are aware, there is no prior precedent.

Fig1 – I find this map very helpful. I would recommend adding the names of the different regions mentioned in the study to the map.

Good suggestion, we have added names of countires and relevant oceans to the map.

144 – why do you think the number of optimal clusters was different for the four AR detection tools? What characteristic exerts the greatest influence in the clustering, and

what does that suggest about the differences among the catalogs?

This is a good question, and also a tricky one. The biggest difference among the different ARDTs is the spatial extent of the ARs that they detect. With the larger ARs, the overlap in the data will inherently be larger, which in turn will aid the clustering. So for the ARDTs detecting smaller ARs, we suspect that more clusters are needed to capture the variability among the ARs. But the resulting clusters are large enough to overlap so that we can combine them.

153 – do you believe that non-AR associated precipitation occurs during an AR event, in the footprint of the AR? Or is this because you're summing all precipitation in Scandinavia during AR landfall anywhere in the region?

Fully attributing the precipitation during an AR to different sources is tricky. We expect that there can be non-AR associated precipitation in the AR footprint. This is why we subtract the climatological average of non-AR precipitation, as this should isolate the AR-specific contribution from the background precipitation.

177-179 – consider moving this to the methods

We see value in having it here to remind the reader of what they are going to look at.

180-185 – what is the standard deviation in the frequency and precipitation?

Since our ensemble only consists of four members, we think that the standard deviation will not be representative/meaningful. Instead, we present the minimum and maximum of the ensemble. We see that this was missing here and have added it (L180–184) – thanks!

191 – add "frequency" > "annual AR frequency and precipitation"

Thanks! We have corrected this (L190).

206 – how did you decide on this cutoff for high precipitation amounts?

We inferred this from panel b in Figure 4. It shows that the probability densities are higher during AR events for precipitation rates 12 mm 6h$^{-1}$ and above. In the revised manuscript, we have reworked this paragraph to not use this cutoff (L205–210).

209 – consider converting to hourly or daily rates, which may be more comparable across different studies, as opposed to mm per 6h

This is a valid point; we do however think that converting from the 6-hourly accumulated values to either mm/h or mm/day could be misleading. The distribution of hourly values is likely different from the 6-hourly values, while for the conversion to daily values we would have to assume that the 6-hourly precipitation persist throughout the day. Alternatively, if we further aggregate the 6-hourly values to daily, it would be nontrivial to assign the daily precipitation to AR events that last for less than a full day (e.g. half a day, or 2.25 days).

Therefore, we think the clearest approach is to work with the 6-hourly precipitation.

Fig5 – would it make sense to put 5a,c,e, and g on a common colorbar (value range for the colors)? And same for b,d,f, and h? Currently the total precipitation fraction for 5f looks much larger than any of the other panels, purely because of the dominance of dark blue colors, even though in reality it has smaller values than the other panels.

We have been going back and forth on this throughout the development of the study. The benefit of having the same colour scale for the subplots is, as you mention, that it would be more straightforward to compare the clusters to each other in terms of absolute magnitudes. However, due to the relatively large range, a lot of (if not all) the details in the less active clusters would be lost in the darker colours. So we have opted to have separate colour scales, and we are trying to make this clear by always referencing the maximum values in the text. To further clarify this, we have added annotations showing the max value for each subplot, as is done in Fig. 8 to all figures showing maps.

Fig6 – similar to the previous comment – you could normalize these so that the colorbar is the same and the percentage of ARs by season sums to 1.

See the response to the comment above. Additionally, we did think about some type of normalization for these plots, but we also wanted the numbers to be easy to interpret. For instance a frequency of 4% here corresponds directly to the fraction of time steps within that season. Any normalization would make the figure less intuitive and require an additional explanation of what is shown.

264-267 – could add these place names to Fig1

Yes, great suggestion. This has been added.

319 – "are typically lower" add "due to lower air temperatures"

Thanks! We have included this (L325).

**3 Response to referee 3**

See https://doi.org/10.5194/egusphere-2025-3992-EC1

We thank referee 3 for taking the time to assess our manuscript and providing valuable feedback. In this document, we have included said feedback along with our responses in blue.

The manuscript studies the climatology of atmospheric rivers (ARs) in Scandinavia. The authors investigate the frequency, seasonality, and associated precipitation of four different pathways through which ARs reach Scandinavia, identified using k-means clustering. Additionally, they examine the relationship between large-scale atmospheric configurations (through the NAO index) and AR occurrences in the region. The manuscript is well written and well structured, and it addresses a relevant topic that has not been previously studied for the Scandinavian region. The manuscript is suitable for publication in WCD after some methodological clarifications.

1. The choice of ARDTs (Section 2.1): The ARTMIP project offers a large number of ARDTs. Why were these four chosen? Regarding the choice of ARDTs, there are other considerations to take into account:

    The four ARDTs were selected because they are the AR catalogues published by ARTMIP that has been applied to ERA5, has a global spatial coverage and span 1980–2019. Other published AR catalogues either use a different reanalysis product, span a shorter time period, or does not cover the area we are interested in.

    We have updated L80–84 to clarify this.

    (a) TempestLR algorithm (Table 1): The threshold used is based on the Laplacian of IVT. The threshold of 250 kg m$^{-1}$s$^{-1}$ stated there is probably not correct (Ullrich et al., 2021). Please make sure to include the correct threshold used in the catalogue.

    You are correct. On the ARTMIP website, the TempestLR algorithm threshold is described as dIVT $\geq$ 250 kg m$^{-1}$s$^{-1}$. But we have changed this in Table 1 to that the Laplacian of the IVT field needs to be $\leq$ -40 000 kg m$^{-1}$ s$^{-1}$ rad$^{-2}$, with a note that this does capture points with IVT $\leq$ 250 kg m$^{-1}$s$^{-1}$.

    (b) Reid500 algorithm: Using a fixed threshold of 500 kg m$^{-1}$s$^{-1}$ does not seem very suitable for Scandinavia, where some regions lie within the polar circle and these values of IVT might be too high. There are ARDTs specifically tailored for ARs in polar regions; perhaps one of those would be more appropriate for this study (Gorodetskaya et al., 2014).

    For this study, we selected all global AR catalogues based on ERA5 and spanning at least 1980–2019 from the ARTMIP project. Given that Reid500 fulfils these criteria, we did not want to exclude it based on prior assumptions.

    Based on our results, we agree that other ARDTs or a lower fixed threshold

are more suitable for the Scandinavia region. We have updated L80–85 to makes this more clear and expanded the discussion on ARDT differences and suitability on L313–336 in the revised manuscript.

(c) Line 324: Uncertainties regarding ARDTs are presented here. It would be nice if the authors could provide a recommendation of which of ARDTs are more suitable for this region, which lies between the mid-latitudes and polar regions, as no ARDTs are specifically designed for it.

Thank you for this suggestion. It is difficult to recommend ARDTs without a thorough evaluation, which is challenging and beyond the scope of our current study. Instead, we use an ensemble approach to quantify the uncertainty that arises from applying different global ARDTs in this region. We have added a brief discussion of which types of ARDTs are likely more suitable for Scandinavia on L331–336 in the revised manuscript.

2. In Lines 105–107, it is described how ARs are tracked, with the tracking condition being "4- connected component labeling". It is not clear what this means. Please expand this explanation to make it clearer. Could it mean that an AR should be detected through four consecutive time steps?

The component labelling algorithm was used to identify and separate individual AR objects ("blobs") within each binary mask. This has been clarified in the revised manuscript on L100–105 in the revised manuscript.

"4-connected" is a common term in image processing. In this scheme, pixels that share an edge—here, meaning they are adjacent in the north, south, east, or west directions—are considered connected. This is just a detail and now appears in parentheses.

3. The AR tracking condition mentioned in the previous comment is applied to the detected blobs from the ARDTs in the catalogues, which themselves do not track ARs, this is not obvious since some of these ARDTs also track ARs. You mention this in Lines 334–336; this should be stated at this point as well.

Thanks, we have clarified the sentence about that the AR catalogues we have used do not contain information about the temporal relationship between AR objects on L113–114 in the revised manuscript.

4. In Line 113, the domain where the AR tracking is applied is introduced. It could be presented earlier (even in the previous Data Section), as this would improve the understanding of the methodology for readers who are not familiar with AR detection and tracking procedures. However, applying the tracking threshold to such a small domain might lead to missing ARs that reach the area for fewer than four time-steps. Considering a larger domain when applying the AR tracking condition could help prevent this issue. In Figure 2a, the Scandinavian AR frequency is shown, with a maximum around the west coast of Denmark, and the frequency

decreases considerably around the British Isles, which typically exhibit a higher AR frequency (Thandlam et al., 2022; Guan et al., 2015) than the North Sea. Please ensure that this pattern is indeed due to the condition that only ARs intersecting Scandinavia are included, and not an artefact of applying the AR-tracking condition within too small a domain.

Thank you for this suggestion. We have made substantial revisions throughout the Methods section to better clarify our methodological approach. First, we do not perform any AR identification ourselves, but rely on global AR catalogues provided by ARTMIP. Second, the steps prior to introducing the domain were carried out on the global data, so we believe it makes more sense to introduce the domain at this point. Third, there was no constraint on the number of time steps—ARs only need to intersect the region (Scandinavia) for a single time step for them to count as landfalling. Fourth, limiting the domain simply means excluding segemts of AR objects that fall outside the specified geographical boundaries, not the entire AR objects. This exclusion does not affect the results presented in our figures.

Regarding your concern about the AR frequency around the British Isles, we confirm that this is indeed due to including only ARs that intersect Scandinavia. Our limited spatial domain does not affect these results.

5. In Line 153, it is explained how the AR precipitation is calculated, which is done by subtracting the non-AR precipitation. However, how is the non-AR precipitation calculated? I assume this is done when computing the field correlations between AR frequency and precipitation patterns. This process is not very clear, please clarify and improve the explanation in this section.

The non-AR precipitation is the precipitation that occurs during time steps when there are no ARs present over the region. In the revised manuscript, we have clarified this on L153.

6. The NAO bins (Lines 168–171) are defined in four groups of equal size centered at 0. Nonetheless, this choice seems somewhat arbitrary and, later in the Results section, appears to limit the interpretation of the results (as all clusters seem to be mostly associated with a "strong positive" NAO). Since the NAO index is used as an indicator of large-scale patterns, the bins defined here might not be representative of specific large-scale configurations. For example, from 0 to 0.5 it is defined as a positive NAO, and between 0.5 and 3.5 as a strong positive NAO, but a value of 0.2 might not be representative of a positive NAO pattern, and a value of 0.7 might not represent a strong positive NAO, whereas a value of 2 likely would. Have you investigated whether your results are sensitive to the choice of these bins? One could consider defining NAO bins that are more physically relevant for certain weather patterns (e.g., Song et al., 2014, use a different threshold for NAO bins) or applying a weather regime approach (e.g., as in Messori et al., 2023), which goes beyond the NAO index.

This is very valuable feedback, thank you. Our initial choice of NAO bins was based on splitting the positive and negative phases in groups of equal size, which resulted in the 0.5 limit between positive (negative) and strong positive (negative) bins. We agree that this is somewhat arbitrary, and that the positive/negative phases include relatively weak phases. In the revised manuscript, we have updated the NAO bin definitions to include the neutral phase (as in Song et al., 2014). We still split the positive and negative groups into two bins, so this resulted in five groups in total: NAO $<$ -1.5, -1.5 $\leq$ NAO $<$ -0.5, -0.5 $\leq$ NAO $\leq$ 0.5, 0.5 $<$ NAO $\leq$ 1.5, and NAO $>$ 1.5. This has not affected the outcome of our analysis—the more northern ARs are still much more frequent during the strong positive phases of the NAO—but nevertheless this is good improvement to our method. Changes relevant to this comment are found on L166–168, and in section 3.4 Influence of NAO on AR pathways in the revised manuscript.